# Why Playing Against Diverse and Challenging Opponents Speeds Up Coevolution: A Theoretical Analysis on Combinatorial Games

**Alistair Benford**
School of Computer Science
University of Birmingham
a.s.benford@bham.ac.uk

**Per Kristian Lehre**
School of Computer Science
University of Birmingham
p.k.lehre@bham.ac.uk

## Abstract

Competitive coevolutionary algorithms (CoEAs) have a natural application to problems that are adversarial or feature strategic interaction. However, there is currently limited theoretical insight into how to avoid pathological behaviour associated with CoEAs. In this paper we use impartial combinatorial games as a challenging domain for CoEAs and provide a corresponding runtime analysis. By analysing how individuals capitalise on the mistakes of their opponents, we prove that the Univariate Marginal Distribution Algorithm finds (with high probability) an optimal strategy for a game called Reciprocal LeadingOnes within $O(n^2 \log^3 n)$ game evaluations, a significant improvement over the best known bound of $O(n^5 \log^2 n)$. Critical to the analysis is the introduction of a novel stabilising operator, the impact of which we study both theoretically and empirically.

## 1 Introduction

Many of the most popular machine learning methods for multi-agent systems rely on self-play. Indeed, recent breakthroughs on games including Go, Chess, and Poker [43, 5, 34] have used self-play to produce superhuman performance without prior training data. However, despite their popularity self-play algorithms are not provably efficient in general, and a robust theory of which self-play algorithms are efficient and how performance scales with problem size is an important open question.

A recurring observation is that while self-play agents perform well against similar play styles, they often perform poorly against new agents exhibiting radically different strategies to those trained against [32, 48, 44]. Accordingly, there is a need to ensure strategic diversity is present during learning. A possible source for this strategic diversity is to utilise population-based approaches such as competitive *coevolutionary algorithms* (CoEAs) [40]. CoEAs iteratively evaluate individuals of a population based on interactions with competitors, selecting the strongest as parents for the next generation's population, which is instantiated using mutation and crossover.

The successful application of CoEAs is not straightforward in general, due to the possibility of pathological behavior such as cycling, forgetting, and loss of fitness signal [12]. These issues are often caused by a failure to accurately ascertain the quality of individuals, which in turn relies on ensuring opponents are both diverse and challenging. For example, a lack of population diversity on highly intransitive games can lead to cycle-chasing; if a certain genotypic feature is needed to defeat a particular type of opponent, that feature may be forgotten if such opponents are not also retained in the population; and evaluations against significantly stronger or weaker opponents provide little useful information for progression, which incurs a loss of fitness signal.

39th Conference on Neural Information Processing Systems (NeurIPS 2025).

As it is critical to avoid such behaviours, our research objective is to gain a deeper understanding of how the distribution of opponents is affected by algorithm design, and how this subsequently impacts the performance of CoEAs. This understanding can be acquired through *runtime analysis*, which provides rigorous estimates of the number of times the game is simulated by an algorithm until a specified objective is achieved [8]. While there is a large amount of runtime analysis for standard evolutionary algorithms (EAs) [9], the first steps towards a corresponding theoretical framework for coevolution have only been taken recently (see Section 1.1), despite clear demand [40].

As theoretical work on full-scale real-world problems is generally not feasible, the problems to which runtime analysis applies are often simplifications designed to capture a challenging aspect of the corresponding application. There is a large amount of interest in the use of CoEAs for game playing (see Section 1.1), so we consider this a critical area for the development of runtime analysis for CoEAs. One important class of difficult and varied games are *combinatorial games*. Even though optimal play for small combinatorial games can be computed using the game graph in linear time (as a function of $n$, the number of game positions), the full space of possible strategies is exponential in $n$, and CoEAs often struggle to find winning strategies efficiently. Thus combinatorial games can provide a challenging class of benchmarks for CoEAs that are amenable to theoretical analysis.

Recently, the first runtime analysis for CoEAs on combinatorial games [4] showed that for any impartial combinatorial game, a CoEA called Tournament UMDA (see Section 2) is likely to find an optimal strategy within $n^{O(\overline{s})}$ game evaluations, where $\overline{s}$ is a precisely defined invariant of the corresponding game graph. For many games, $\overline{s}$ turns out to be a constant. However, even in such cases, the implied exponent in the polynomial bound is much higher than implied by observed performance, and it was conjectured that a more sophisticated analysis of how competing individuals respond to the weaknesses of their opponents could yield tighter runtime bounds.

The main contribution of this paper is to advance the development of a theoretical framework for coevolution by undertaking this more detailed analysis. We will introduce a combinatorial game called Reciprocal LeadingOnes (RLO) which is difficult for a range of CoEAs and for which the aforementioned analysis implies a runtime bound of $O(n^5 \log^2 n)$. With careful examination of the corresponding coevolutionary dynamics, we prove the following improved upper bound, as well as a new lower bound demonstrating the result is tight up to a polylogarithmic factor.

**Theorem 1.1** (Theorem 5.2, informal version). *With appropriate parameters, Tournament UMDA finds with high probability an optimal strategy for* RLO *in time bounded above by* $O(n^2 \log^3 n)$ *and bounded below by* $\Omega(n^2 \log n)$.

The particular variant of UMDA we analyse includes a novel feature that promotes diversity in the resulting populations. Not only does its inclusion greatly assist the theoretical analysis, but we will see in the experimental analysis of Section 4 that it also improves the algorithm's observed performance. Our experiments will also investigate the performance of a range of CoEAs on several combinatorial games, and help motivate RLO as a challenging benchmark for CoEAs.

While population diversity is an important requirement for success on RLO, diversity alone cannot overcome its challenges. Indeed, we will also prove that any evolutionary algorithm (EA) that maximises diversity by foregoing coevolution and evaluating only against uniform random opponents has exponential runtime on RLO, thus confirming that a coevolutionary arms race between players and opponents is essential for sustained progress towards optimal play.

**Theorem 1.2** (Theorem 6.1, informal version). *If an EA is used to optimise* RLO, *where evaluations are made by playing against uniform random opponents, then with overwhelming probability, exponential time is needed to discover an optimal strategy.*

## 1.1 Related work

There have been numerous case studies on the topic of coevolution for combinatorial games, including for Nim and Tic-Tac-Toe [41, 23], Backgammon [39], Othello [24, 44, 45], Senet [11], Checkers [6], Chess [13, 18], and Go [31]. More general game-playing applications of coevolution include Pong [33], Bomberman [15], Poker [36], and Resistance [26]. Notably, DeepMind's groundbreaking AlphaStar algorithm made use of a competitive coevolutionary league training system [48, 1].

The first runtime analysis for competitive CoEAs was provided by [27], which established conditions for the successful location of the Nash equilibrium of an intransitive game called BILINEAR by

a population-based CoEA. Further theoretical analysis of CoEAs on BILINEAR has studied the impact of algorithmic features including archives [21] and fitness aggregation [19]. In this paper, we ask how to retain a distribution of diverse opponents that are conducive to coevolutionary learning. This question was also examined theoretically in [20] through a method promoting populations of individuals that are diverse in how they rank opponents relative to one another. The coevolutionary instance of UMDA to which our main result applies was first analysed on a class of symmetric zero-sum games [3], and also later in the first runtime analysis of a coevolutionary algorithm for combinatorial games (and indeed, any turn-based game) [4].

## 1.2 Notation

Let $S$ be a finite set. A *probability distribution over $S$* is a function $p : S \to [0, 1]$ such that $\sum_{s \in S} p(s) = 1$. An $S$-valued random variable $x$ is said to be *distributed according to $p$*, denoted $x \sim p$, if $\mathbb{P}(x = s) = p(s)$ for all $s \in S$. For any subset $A \subseteq S$, we define $p(A) = \sum_{s \in A} p(s)$. Given $\gamma \in [0, 1]$, let $\mathcal{P}_\gamma(S)$ be the set of distributions $p$ over $S$ with $p(s) \geqslant \gamma$ for all $s \in S$, and define $\mathcal{P}(S) = \mathcal{P}_0(S)$. We use $\mathscr{P}(S)$ to denote the powerset of a set $S$. All logarithms are the natural logarithm unless stated otherwise, and given $k \in \mathbb{N}$ we write $\log^k n = (\log n)^k$.

## 2 Tournament UMDA

Whereas most EAs generate new individuals by mutating existing individuals in the current population, *estimation of distribution algorithms* (EDAs) instead use statistics about the current population to instantiate a more general probability distribution over the entire search space [38]. Candidates for selection in the next generation are then sampled according to this distribution.

For search domains of the form $\mathcal{X} = \prod_{i \in I} S_i$, where $I$ is a finite indexing set, a standard practice is to keep track of a *frequency vector $p_t \in \prod_{i \in I} \mathcal{P}(S_i)$* that evolves with time. Broadly speaking, $p_t(i)(s)$ represents the proportion of individuals in the population $P_t \subseteq \mathcal{X}$ that have an $s$ appearing in position $i$. New search points can then be generated using the following probability distribution.

**Definition 2.1.** *Given $\mathcal{X} = \prod_{i \in I} S_i$ and $p \in \prod_{i \in I} \mathcal{P}(S_i)$, let $\mathrm{Univ}(\mathcal{X}, p)$ denote the probability distribution over $\mathcal{X}$ such that, if $x \sim \mathrm{Univ}(\mathcal{X}, p)$, then for any $y \in \mathcal{X}$, $\mathbb{P}(x = y) = \prod_{i \in I} p(i)(y)$.*

The coevolutionary EDA we analyse is a one-population instance of the Univariate Marginal Distribution Algorithm (UMDA) which employs a binary tournament selection, and is delineated in Algorithm 1 (for further comparison of one-population and two-population CoEAs, see Section 4).

---

**Algorithm 1** Tournament UMDA with parameters $\mu \in \mathbb{N}$, $\gamma > 0$, $c > 0$

---

**Require:** Search domain $\mathcal{X} = \prod_{i \in I} S_i$ and payoff function $f : \mathcal{X} \times \mathcal{X} \to \{-1, 1\}$.
  **for** $i \in I$ and $s \in S_i$ **do**
    Set $p_0(i)(s) = \frac{1}{|S_i|}$.
  **end for**
  **for** $t \in \mathbb{N}$ until termination criterion met **do**
    **for** $j \in [\mu]$ **do**
      Sample $x \sim \mathrm{Univ}(\mathcal{X}, p_t)$ and $y \sim \mathrm{Univ}(\mathcal{X}, p_t)$
      Set $w = \begin{cases} x & \text{if } f(x, y) = 1 \\ y & \text{if } f(x, y) = -1 \end{cases}$
      Sample $P_{t+1}(j) \sim \mathrm{MUTATE}(c, p_t, w)$
    **end for**
    **for** $i \in I$ **do**
      **for** $s \in S_i$ **do**
        Set $q_{t+1}(i)(s) = \frac{1}{\mu} |\{j : P_{t+1}(j)(i) = s\}|$
      **end for**
      Set $p_{t+1}(i) = \pi_\gamma^{S_i}(q_{t+1}(i))$
    **end for**
  **end for**

---

UMDA was first introduced as a non-coevolutionary EDA by Mühlenbein and Paaß [35], and the instance analysed here is closely related to that appearing in [3].

As is common for EDAs, there is a special step at the end of each generation to ensure that the probability of a generated individual taking any given value $s \in S_i$ in position $i \in I$ never drops below some boundary parameter $\gamma$. This is achieved by use of constraint functions $\pi_\gamma^{S_i} : \mathcal{P}(S_i) \to \mathcal{P}_\gamma(S_i)$. We refer the reader to [4, 2] for a complete description of $\pi_\gamma^{S_i}$. However, we note that for our proofs we only rely on the fact that the constraint function simplifies to $\pi_\gamma^{S_i}(p)(s) = \min\{\max\{p(s), \gamma\}, 1 - \gamma\}$ in the case $|S_i| = 2$. A novel feature of the instance of UMDA described here (and indeed the only difference to that appearing in [4]) is the use of an operator $\text{MUTATE}(c, p_t, x)$ applied to each selected individual.

**Definition 2.2.** *Given $x \in \mathcal{X} := \prod_{i \in I} S_i$, the operator $\text{MUTATE}(c, p_t, x)$ generates an element $y \in \mathcal{X}$ by, independently for each $i \in I$, sampling $y(i)$ uniformly at random from $S_i$ with probability $c \cdot \prod_{s \in S_i}(1 - p_t(i)(s))$, and otherwise setting $y(i) = x(i)$.*

$\text{MUTATE}(c, p_t, x)$ introduces a probability that individuals entering the next population have some entries generated uniformly at random, thus nudging each entry $p_t(i)$ of the frequency vector slightly towards the uniform distribution. This increases the level of population diversity in subsequent generations. Nonetheless, because the size of the nudge is proportional to $\prod_{s \in S_i}(1 - p_t(i)(s))$, its effect lessens when $p_t(i)$ has low entropy, and thus does not prevent the algorithm converging on those positions $i \in I$ where there is a strong preference for certain values.

As well as its potential diversity benefits, the inclusion of $\text{MUTATE}(c, p_t, x)$ is perhaps best motivated in terms of its relation to the concepts of balance and stability as applied to EDAs [14, 25]. Roughly speaking, an EDA is *balanced* if $\mathbb{E}[p_{t+1} \mid p_t] = p_t$ holds in the absence of any fitness signal, and it is *stable* if the distributions $(p_t(i))_{i \in I}$ remain close to uniform in such a case. While these properties do not seem at first to be obviously contradictory, Theorem 6.11 of [25] shows that they are in fact mutually exclusive. In the same chapter, it is shown via an application to the well-known LEADINGONES benchmark (where the fitness of a bitstring is defined as the length of its longest prefix of 1-bits) that stable EDAs can offer significant speed up due to their prevention of premature convergence on positions that do not become relevant until later in the optimisation process.

The addition of the operator $\text{MUTATE}(c, p_t, x)$ transforms Tournament UMDA from a coevolutionary EDA that is balanced into one that is stable. With this, we should hope to see improved performance in settings where the optimisation of positions happens sequentially rather than concurrently, as is the case for LEADINGONES. We will see in Section 4 that this change significantly improves the performance of Tournament UMDA on combinatorial games, and also assists with the proof of the associated runtime bounds.

## 3 Impartial combinatorial games and Reciprocal LeadingOnes

We quickly describe the representation of impartial combinatorial games via directed graphs; for a more comprehensive introduction, see [4, 16, 17]. An *impartial combinatorial game* (ICG) is a finite acyclic rooted directed graph $G = (V, F, v_0)$, where $V$ is a vertex set of size $n$ (the *game positions*), $v_0 \in V$ is the initial game position, and $F : V \to \mathscr{P}(V)$ is a function mapping each position onto those which can be reached in a single move. Players take turns moving the current position $v$ to an element of $F(v)$. If $F(v)$ is empty, the current player loses.

We will encode strategies for ICGs as an assignment of each interior vertex $v$ to an element of $F(v)$, with this assignment indicating the preferred move at each game position. Formally, using $\text{Int}(G)$ to denote the set of $v \in V$ with $F(v) \neq \emptyset$, then $\mathcal{X}_G = \prod_{v \in \text{Int}(G)} F(v)$ will be the set of strategies for $G$. Given $x \in \mathcal{X}_G$, let $x(u) \in V$ denote the preferred move at at position $u$. Let $f_G : \mathcal{X}_G \times \mathcal{X}_G \to \{-1, 1\}$ be the payoff function for $G$, where $f_G(x, y) = 1$ indicates that $x$ wins against $y$ and $f_G(x, y) = -1$ indicates that $x$ loses against $y$ (where $x$ moves first).

To help motivate Reciprocal LeadingOnes, let us consider why learning winning strategies even for simple combinatorial games can be difficult in principle. Whether it is Chess or Nim, in order to guarantee optimal play in a combinatorial game it is not enough to put oneself in a winning position; one must then also maintain the upper hand by responding correctly to any difficult position an opponent move the game to later. However, for many combinatorial games, each play only bears

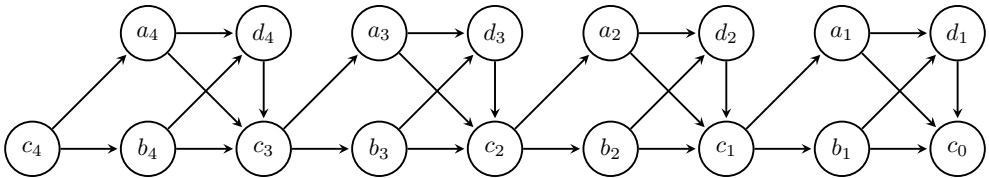

Figure 1: The game graph $G$ for RLO.

witness to a fraction of the overall possible positions. It is on such games that learning is potentially improved by training against opponents that are both challenging (so that novel and difficult positions occur) and diverse (so that a range of difficult positions occur).

We define Reciprocal LeadingOnes (RLO) to be the impartial combinatorial game $G = (V, F, v_0)$ with the specification $V = \{c_0\} \cup \left( \cup_{i \in [n]} \{a_i, b_i, c_i, d_i\} \right)$, $v_0 = c_n$, and

$$F(v) = \begin{cases} \{c_{i-1}, d_i\} & \text{if } v = a_i \text{ or } v = b_i, \\ \{c_{i-1}\} & \text{if } v = d_i, \\ \{a_i, b_i\} & \text{if } v = c_i \text{ and } i > 1, \\ \emptyset & \text{if } v = c_0. \end{cases}$$

An illustration of the game is shown in Figure 1. RLO can be regarded as a series of $n$ challenges and responses, where the player who plays at $c_i$ is considered to be the current challenger and the player who plays at $\{a_i, b_i\}$ is the current responder. In each play of the game only one of each $a_i$ or $b_i$ can be encountered, yet optimal play necessitates perfect play at both, and so RLO captures the notion of a difficult game to learn as described above. As the player who plays at $c_0$ loses, the responder can win by repeatedly moving the game to $c_{i-1}$ (a "correct response"), thus maintaining their position as responder. Thus, RLO is a second-player-win game, and victory can be guaranteed by ensuring that $x(u) = c_{i-1}$ for every $i \in [n]$ and $u \in \{a_i, b_i\}$. In fact, this condition for perfect play is also a necessary one, and so we have the following characterisation of optimal strategies for RLO (see Appendix A for a corresponding proof).

**Definition 3.1.** *Let* $\mathrm{Opt}(G) \subseteq \mathcal{X}_G$ *be the set of* $x$ *such that* $f_G(y, x) = -1$ *for every* $y \in \mathcal{X}_G$.

**Proposition 3.2.** $x \in \mathrm{Opt}(G)$ *if and only if* $x(u) = c_{i-1}$ *for every* $i \in [n]$ *and* $u \in \{a_i, b_i\}$.

## 4 Experimental analysis

In this section we present experimental analysis showing combinatorial games to be a challenging benchmark and case study for CoEAs. Throughout, the runtimes for an algorithm $\mathcal{A}$ on a game $G$ were obtained by performing 100 runs of $\mathcal{A}$ for each population size in a range of values between 100 and 10000, and retaining only the 100 runs associated to the population size with the lowest average runtime. When we refer to the average runtime of $\mathcal{A}$ on $G$, we refer specifically to the mean of these 100 retained runs. All algorithms were run until either the population contained an optimal strategy or $10^8$ game evaluations had been performed, whichever occurred first. As such, runtimes of $10^8$ should be interpreted as failure to discover an optimal strategy in the allocated time. In addition to RLO, we consider a number of other well-known ICGs. Many of the algorithms selected were chosen due to a previous application to combinatorial games (including Chess, Checkers, and Othello). For further details on the algorithms and games, as well as details of computational resources used to carry out the relevant experiments, see Appendix B.

Most CoEAs are two-population, and evaluate individuals based on interactions with the opposing population. For asymmetric problems (especially those with differing search domains), such algorithms are the only viable option. However, for settings where both players have the same strategy space, one may opt to use a one-population CoEA in which individuals are evaluated based on their interactions with other members of the same population. There are potential benefits to this approach, such as to avoid a loss of fitness signal arising from differing strengths between populations. However, reliable insight into when each paradigm should be employed is limited.

The distributions of runtimes shown in Figure 2 relate to three broad classes of CoEAs: one-population EDA-type CoEAs (Tournament UMDA with $c = 0$ and $c > 0$); one-population non-EDA CoEAs

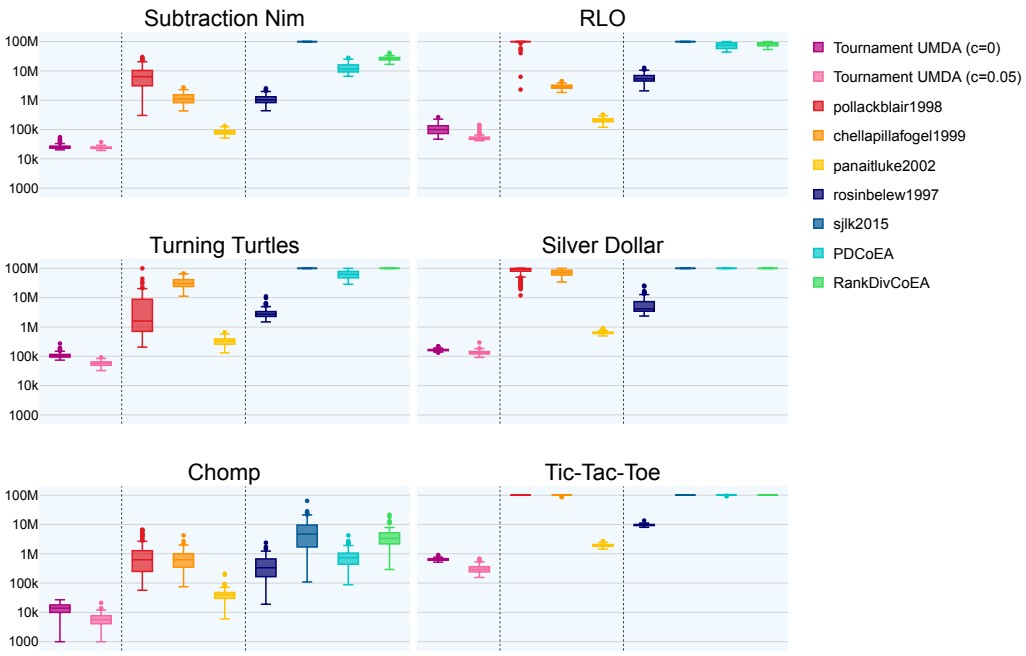

Figure 2: Distributions of runtimes for a range of CoEAs on six combinatorial games. For each algorithm-game combination, the distribution plotted is that of the 100-run batch corresponding to the population size with the fastest average runtime.

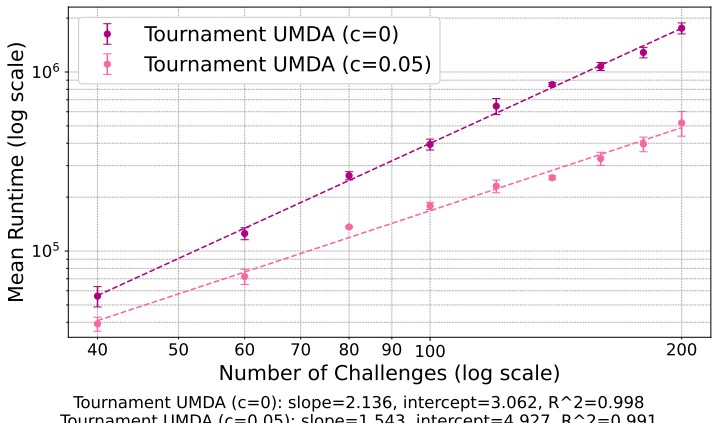

Tournament UMDA (c=0): slope=2.136, intercept=3.062, R^2=0.998
Tournament UMDA (c=0.05): slope=1.543, intercept=4.927, R^2=0.991

Figure 3: Average runtime of Tournament UMDA on RLO with varying number of challenges. Similar to the results shown in Figure 2, runtime was obtained following a basic parameter sweep. That is, for each $n$ batches of 100 runs on RLO of both versions of Tournament UMDA were executed using a range of population sizes. Each mean runtime and 95% confidence interval ($\pm 1.96 \times$ SEM) is that corresponding to batch with lowest average runtime. Dashed lines indicate log–log linear regressions.

Table 1: Comparison of runtimes for Tournament UMDA for $c = 0$ and $c = 0.05$.

| Problem | Tournament UMDA (c=0) | | Tournament UMDA (c=0.05) | | $t$-statistic | $p$-value |
| | Mean Runtime | Std Dev | Mean Runtime | Std Dev | | |
| --- | --- | --- | --- | --- | --- | --- |
| Subtraction Nim | 2.60 e4 | 5.29 e3 | 2.40 e4 | 2.40 e3 | 3.45 | 7.33 e−4 |
| RLO | 1.06 e5 | 4.14 e4 | 5.53 e4 | 1.81 e4 | 11.23 | 4.25 e−21 |
| Turning Turtles | 1.09 e5 | 2.73 e4 | 5.76 e4 | 1.13 e4 | 17.34 | 10 e−35 |
| Silver Dollar | 1.65 e5 | 1.50 e4 | 1.34 e5 | 2.45 e4 | 10.50 | 4.63 e−20 |
| Chomp | 1.38 e4 | 5.82 e3 | 6.20 e3 | 3.02 e3 | 11.54 | 2.12 e−22 |
| Tic-Tac-Toe | 6.48 e5 | 6.48 e4 | 3.13 e5 | 9.76 e4 | 28.35 | 8.86 e−67 |

(pollackblair1998, chellapillafogel1999, and panaitluke2002); and two-population non-EDA CoEAs (rosinbelew1997, sjlk2015, PDCoEA, and RankDivCoEA). Despite the games having relatively small game graphs (ranging from between 64 and 629 vertices), most CoEAs required millions of simulations to discover optimal play (if optimal play was found at all). Notably, the best-performing CoEAs in each of the non-EDA classes (panaitluke2002 and rosinbelew1997) both used crossover and archives, which increase population diversity and help prevent coevolutionary forgetting.

Despite its simplicity compared to the best-performing non-EDAs, the instances of Tournament UMDA were the best-performing algorithms on all games, justifying its selection as an interesting case for further theoretical analysis. Notably, setting the parameter $c$ to be positive in the mutation operator $\text{MUTATE}(c, p_t, x)$ improved the performance of Tournament UMDA on all six games. Table 1 compares the relative performance for $c = 0$ and $c = 0.05$, and the speed up was found to be significant with respect to Welch's $t$-test [49]. As these findings only apply to problems of a fixed size, we also present in Figure 3 an analysis of how the runtime of Tournament UMDA scales with the number of challenges $n$ in RLO. Polynomial performance is observed from the linear relationship between $n$ and runtime when plotted on a log-log scale. It is remarkable that while the estimated degree of the polynomial for $c = 0$ closely corresponds to the dominant term of $n^2$ in our main theoretical result, the slope of 1.543 for $c = 0.05$ indicates that asymptotically superior performance may be possible as a result of introducing $\text{MUTATE}(c, p_t, x)$ for populations of size roughly $\sqrt{n}$ (that is, smaller than considered in our analysis).

## 5 An upper bound for the runtime of UMDA on Reciprocal LeadingOnes

The runtime of a CoEA is measured as the number of times the payoff is queried until the algorithm reaches the desired search objective (see [10]), as follows.

**Definition 5.1.** *Suppose that $G$ is an impartial combinatorial game, and that $\mathcal{A}$ is an algorithm which makes $\tau$ queries of $f_G$ during each generation. Then, given a set $B \subseteq \mathcal{X}_G$, the* runtime *of $\mathcal{A}$ on $f_G$ is defined to be the random variable $T_{\mathcal{A}}^G(B) = \tau \cdot \min\{t : P_t \cap B \neq \emptyset\}$, where $P_t \subseteq \mathcal{X}_G$ is the population of $\mathcal{A}$ at the start of generation $t$.*

As discussed in Section 1, an upper runtime bound of $O(n^5 \log^2 n)$ for Algorithm 1 on RLO can be quickly obtained using the results of [4] (see Appendix C.1). Our main result improves this upper bound and further establishes a lower bound, as follows.

**Theorem 5.2.** *Let $K > 0$ be an arbitrary constant, let $G$ be the game graph for RLO, and let $\mathcal{A}$ be described by Algorithm 1 where $c \leqslant 1/600$ is a constant and $\gamma = 1/(100n)$. Then there exists $n_0 \in \mathbb{N}$ such that, provided $n \geqslant n_0$ and $\mu \geqslant \left(\frac{(K+4)10^5}{c^2}\right) n \log n$,*

$$\mathbb{P}\left[n^2 \log n \leqslant T_{\mathcal{A}}^G(\text{Opt}(G)) \leqslant \mu n \log^2 n\right] \geqslant 1 - n^{-K}.$$

While the full proof of Theorem 5.2 is provided in Appendix D, here we provide a description of some of the key ideas behind the upper bound, and how they relate to the impact of the operator $\text{MUTATE}(c, p_t, x)$.

A run of $\mathcal{A}$ on RLO generates a Markov chain $(p_t)_{i=0}^{\infty}$ over a state space which we label $\mathcal{Q}$. For each $0 \leqslant i \leqslant n$, we will let $A_i$ be the set of $p \in \mathcal{Q}$ such that $p(c_j, a_j) \in [\frac{1}{8}, \frac{7}{8}]$, $p(a_j, c_{j-1}) \geqslant 1 - 2\gamma$, and $p(a_j, c_{j-1}) \geqslant 1 - 2\gamma$ whenever $j \in [i]$ (where we use $p(u, v)$ as a shorthand for $p(u)(v)$). Note that $\mathcal{Q} = A_0 \supseteq A_1 \supseteq \ldots \supseteq A_n$. The intuition here is that $p_t \in A_i$ corresponds to evaluating

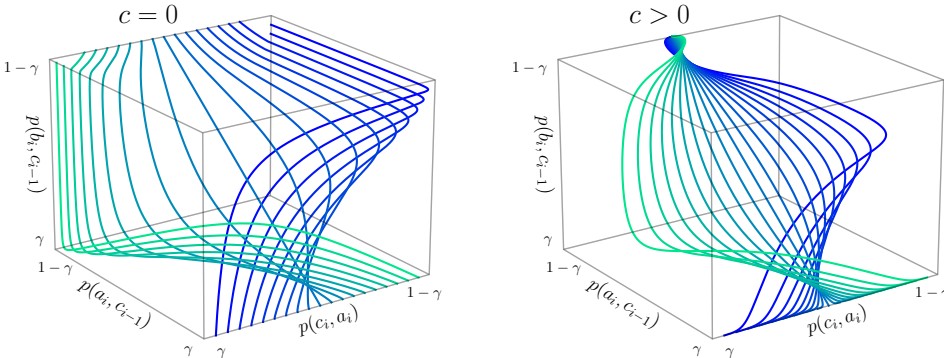

Figure 4: Deterministic trajectories in $[\gamma, 1-\gamma]^3$ induced by $\Delta(\mathbf{q})$ initialised on the line $p(a_i, c_{i-1}) = p(b_i, c_{i-1}) = \gamma$. The cases $c = 0$ (left) and $c > 0$ (right) are both shown. As $\Delta(\mathbf{q})$ is an illustrative device and does not explicitly feature in the proof of Theorem 5.2, the expression for $\Delta(\mathbf{q})$ used to produce this plot (via Euler's method) is one corresponding to the lower bounds featuring in the definition of $\phi_\alpha$, as given in Appendix D.

against opponents that are both diverse and strong in the final $i$ challenges. Indeed, if $\gamma$ is small and $x \sim \mathrm{Univ}(\mathcal{X}_G, p_t)$ for some $p_t \in A_i$, then $x$ is likely to always respond correctly in the final $i$ challenges, making $x$ impossible to beat when entering those challenges as responder. Moreover, because $p_t(c_j, a_j), p_t(c_j, b_j) \in [\frac{1}{8}, \frac{7}{8}]$ for $j \in [i]$, if $x$ enters those challenges as challenger, then $x$ will discover and punish suboptimal opponents with at least constant probability. The diversity condition $p_t(c_j, a_j) \in [\frac{1}{8}, \frac{7}{8}]$ is critical to sustaining consistent progress by ensuring suboptimal players do not reenter the population (as observed in Appendix C.2).

If $x \sim \mathrm{Univ}(\mathcal{X}_G, p)$ for some $p \in A_n$, then $x$ is likely to belong to $\mathrm{Opt}(G)$. Thus, we analyse how long it takes $(p_t)_{t=0}^{\infty}$ to move from $A_0$ to $A_n$. This process is partitioned into $n$ phases, with $p_t$ moving from $A_{i-1}$ to $A_i$ in phase $i$. More specifically, phase $i$ examines how the vector

$$\mathbf{q}_t := (p_t(c_i, a_i), p_t(a_i, c_{i-1}), p_t(b_i, c_{i-1})) \in [\gamma, 1-\gamma]^3$$

moves towards the set $\hat{A} := [\frac{1}{8}, \frac{7}{8}] \times [1-2\gamma, 1-\gamma]^2$. Applying results from [4], and also carefully accounting for the effects of $\mathrm{MUTATE}(c, p_t, x)$ and the constraint functions, we will be able to obtain a fairly precise estimate of the value of

$$\Delta(\mathbf{q}) := \mathbb{E}[\mathbf{q}_{t+1} - \mathbf{q}_t \mid \mathbf{q}_t = \mathbf{q}, \, p_t \in A_{i-1}].$$

$\Delta(\mathbf{q})$ defines a (deterministic) vector field over $[\gamma, 1 - \gamma]^3$. In this framing, our analysis reduces to:

(A) Show the trajectories induced by $\Delta(\mathbf{q})$ converge to $\hat{A}$.

(B) Show the random process $(\mathbf{q}_t)_{t=0}^{\infty}$ closely follows these trajectories.

These tasks are the most detailed and technical and part of our proof. However, they are greatly assisted by the role played by $\mathrm{MUTATE}(c, p_t, x)$. To illustrate, consider Figure 4, which shows these trajectories for the cases $c = 0$ and $c > 0$. When $p_t \in A_{i-1}$, the winner $w$ sampled in Algorithm 1 is typically whichever of $x$ or $y$ avoids playing at position $c_{i-1}$. This observation provides strong intuition for the shape of the trajectories. For example, when $p_t(a_i, c_{i-1})$ is large and $p_t(c_i, a_i)$ is small, most games will visit $b_i$ in which instance the responder can win by moving to $c_{i-1}$, thus incurring a strong pressure for $p_t(b_i, c_{i-1})$ to increase in that region of $[\gamma, 1 - \gamma]^3$. As another example, if $p_t(b_i, c_{i-1})$ is large but $p_t(a_i, c_{i-1})$ is small, the challenger is more likely to win if they select $a_i$, thus incurring a strong pressure for $p_t(c_i, a_i)$ to increase. In both cases, the resulting increase is a consequence of rewarding one player for exploiting sub-optimal play exhibited by another. Thus, the shapes of the curves in Figure 4 are representative of how coevolution learns through a feedback mechanism of individuals capitalising on the mistakes of their opponents.

Transferring this intuition to a rigorous confirmation of (A) and (B) is not straightforward, as random variation in $(\mathbf{q}_t)_{t=0}^{\infty}$ makes it unlikely to follow any single trajectory. For one-dimensional stochastic

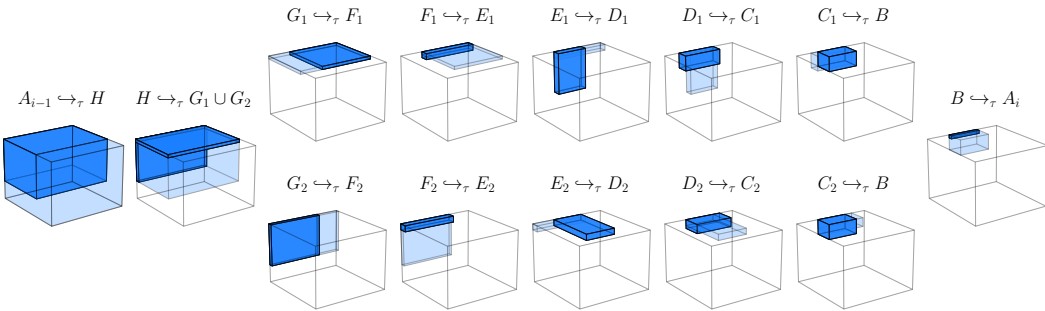

Figure 5: Trajectory-following sequences of cuboids appearing in the proof of Theorem 5.2 (or more precisely, in the proof of Lemma D.8). The three axes of each representation of $[\gamma, 1-\gamma]^3$ correspond to $p(c_i, a_i)$, $p(a_i, c_{i-1})$, and $p(b_i, c_{i-1})$ in the same orientation as in Figure 4. The space close to the boundaries of $[\gamma, 1-\gamma]^3$ has been enlarged to assist with visualisation.

processes, this is typically handled using drift analysis [29]. One way such methods can be applied to higher dimensional processes is by mapping the space down to one-dimension using a distance function, however such a function with the specific properties needed to apply drift analysis is not always guaranteed to exist (and we were unable to find one in this instance). Aside from possibly using a distance function, there is no clear system for generalising drift analysis to higher dimensions beyond simple linear cases [42, 22]. An alternative approach is to use level-based analysis [7, 27], however it was not immediately clear in this case how to define associated levels that fully capture the dynamics of $(\mathbf{q}_t)_{t=0}^{\infty}$. Instead, to handle this technicality, we will introduce a relation $\hookrightarrow_\tau$ between sub-cuboids of $[\gamma, 1-\gamma]$ with the property that if $\mathbf{q}_t \in B$ and $B \hookrightarrow_\tau C$, then $\mathbf{q}_{t+\tau} \in C$ holds with high probability (where we assign $\tau$ a fixed value that is $\Theta(\log^2 n)$). We then define sequences of cuboids, with $\hookrightarrow_\tau$ holding between one cuboid to the next. These sequences collectively follow the paths of the trajectories, and gradually contract down to the set $\hat{A}$. An illustration of this part of the proof is shown in Figure 5.

This transfer from the idealised deterministic progression of Figure 4 to the probabilistic progression of Figure 5 is where the effect of MUTATE$(c, p_t, x)$ is most beneficial. The resulting nudge towards the uniform distribution (recall Section 2) biases the $(\mathbf{q}_t)_{t=0}^{\infty}$ towards the middle of $[\gamma, 1-\gamma]^3$, where the trajectories move more directly towards $\hat{A}$ and where fluctuations caused by random variation are less erratic. This difference is evident also in Figure 4, where the trajectories for $c > 0$ move more directly and smoothly towards $\hat{A}$. This analysis also helps explain the difference in performance between $c = 0$ and $c = 0.05$ observed in Section 4. Accordingly, it is natural to ask how closely the actual trajectories for a run of Tournament UMDA on RLO match the idealisations of Figure 4. For this, we refer the reader to Appendix C.2.

It is useful to consider the extent to which the proof of Theorem 5.2 generalises beyond RLO. The overall proof technique (that is, analysing the trajectory of the vector $\mathbf{q}$ and carrying out steps (A) and (B)) is certainly not exclusive to the specific structure of RLO and there is no innate reason it could not be applied to other combinatorial games. In particular, the proof generalises easily to non-binary cases (that is, combinatorial games with more than two moves available in a turn), albeit with some heavier notation. However, a focus on RLO greatly simplifies our analysis due to its repeating gadget structure, and most of the proof analyses dynamics on a single gadget. Similar arguments could be recreated for game structures that are more complex or varied, however this may be impractical given the high level of technicality already present for RLO.

## 6 EAs cannot optimise combinatorial games by playing against random opponents

The result and corresponding proof of Section 5 helps demonstrate and formalise the benefit provided by evaluating against a range of diverse opponents. However, it would be misguided to conclude that population diversity is the sole contributor to the success of a CoEA – the coevolutionary feedback

between player and opponent is fundamental to learning. To this end, in this section we will show that an evolutionary algorithm cannot efficiently optimise Reciprocal LeadingOnes if individuals are evaluated against opponents sampled uniformly at random from the search domain. In this regime, the distribution of opponents is maximally diverse, but representative of a fixed environment.

The statement and corresponding proof of this black box result relies on the introduction of a family of ICGs $\{\text{RLO}(z) : z \in \{0,1\}^n\}$. Each instance $\text{RLO}(z)$ will be obtainable from RLO by simply relabeling vertices, and so are all isomorphic to RLO. In addition, as a strategy for RLO may be described using $3n$ binary choices, we will adopt $\mathcal{X} := \{0,1\}^{3n}$ as the search domain for all instances of $\text{RLO}(z)$ and write the corresponding payoff function as $f_{\text{RLO}(z)} : \mathcal{X} \times \mathcal{X} \to \mathbb{R}$. For a full description of $\text{RLO}(z)$ and the correspondence of strategies to $\{0,1\}^{3n}$, we refer the reader to Appendix E.1.

---

**Algorithm 2** Black box evolutionary algorithm

---

**Require:** Initial distribution $d_0 \in \mathcal{P}(\mathcal{X})$ and variation functions $v_t : \mathcal{X}^t \times \mathbb{R}^t \to \mathcal{P}(\mathcal{X})$ for $t \in \mathbb{N}$

  Sample $x_0 \sim d_0$
  **for** $t \in \mathbb{N}$ until termination criterion met **do**
    Sample $a_t \sim f(x_t)$
    Sample $x_{t+1} \sim v_{t+1}(x_0, \ldots, x_t; a_0, \ldots, a_t)$
  **end for**

---

The lower bound will apply to any evolutionary algorithm describable by the black box model of [10] outlined by Algorithm 2, which, here, has been generalised to apply to stochastic payoff functions $f : \mathcal{X} \to \mathcal{P}(\mathbb{R})$. We remark that this model is general enough to encompass non-coevolutionary EDAs in addition to classical EAs. With this, we can now state the main result for this section (for a proof, see Appendix E.2).

**Theorem 6.1.** *Given* $z \in \{0,1\}^n$ *and* $x \in \mathcal{X}$, *let* $g_z(x)$ *be the random variable defined by sampling* $y \sim \text{Unif}(\mathcal{X})$ *and setting* $g_z(x) = f_{\text{RLO}(z)}(y, x)$. *Suppose* $\mathcal{A}$ *is any EA describable by Algorithm 2. Then there exists* $z \in \{0,1\}^n$ *such that*

$$\mathbb{P}[T_{\mathcal{A}}^{g_z}(\text{Opt}(\text{RLO}(z))) < 2^{n/8}] \leqslant 2^{-n/8}.$$

It is often standard to restrict black box analysis to algorithms which sample initial search points uniformly at random, and also restrict variation operators to those which are *unbiased*, in the sense that bit values and positions are treated impartially [28]. With these restrictions, the conclusion of Theorem 6.1 holds for *all* instances of Reciprocal LeadingOnes, as follows (for further details on the unbiasedness condition and a proof of Corollary 6.2, we refer the reader to Appendix E.3).

**Corollary 6.2.** *Suppose* $\mathcal{A}$ *is an EA describable by Algorithm 2 which employs the uniform distribution over* $\mathcal{X}$ *as* $d_0$ *and employs only unbiased variation functions* $v_t : \mathcal{X}^t \times \mathcal{R}^t \to \mathcal{P}(\mathcal{X})$. *Then for any* $z \in \{0,1\}^n$ *it holds that*

$$\mathbb{P}[T_{\mathcal{A}}^{g_z}(\text{Opt}(\text{RLO}(z))) < 2^{n/8}] \leqslant 2^{-n/8}.$$

## 7 Concluding remarks

In order to analyse what distributions of opponents are essential to sustain progress in coevolution, we have proven that with high probability a coevolutionary instance of UMDA is able to find an optimal strategy for RLO within $O(n^2 \log^3 n)$ game simulations. Central to this is detailed analysis of a novel mutation operator applied to selected individuals. Experiments indicate that the inclusion of this operator significantly improves the performance of UMDA, which is already among the best-performing CoEAs on combinatorial game benchmarks. We additionally showed that RLO cannot be optimised efficiently by a non-coevolutionary EA evaluating against random opponents.

While our main result provides a lower bound for populations of size $\Omega(n \log n)$ that matches the upper bound (up to a polylogarithmic factor), our experiments indicates that asymptotically faster runtimes may be possible for smaller populations. Related analysis of EDAs indicates that tight bounds for smaller populations may be obtainable, but only with detailed arguments that vary significantly over several cases (see Section 9 of [25]), and so we leave this as an open question for potential future work.

## Acknowledgements

This research was supported by a Turing AI Fellowship (EPSRC grant ref EP/V025562/1). The computations were performed using the University of Birmingham's BlueBEAR HPC service. See `http://www.birmingham.ac.uk/bear` for more details.

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

# A  Proof of Proposition 3.2

*Proof of Proposition 3.2.* Let

$$A = \{x : x(u) = c_{i-1} \text{ for every } i \in [n] \text{ and } u \in \{a_i, b_i\}\}.$$

We have already seen in the discussion of Section 3 that $A \subseteq \mathrm{Opt}(G)$. Now suppose that $x \notin A$, so that there is some $i \in [n]$ and $u \in \{a_i, b_i\}$ such that $x(u) \neq d_i$. If $y \in A$ is such that $y(c_i) = u$, then $f_G(y, x) = 1$ (indeed, if $y$ has not become the responder before the game reaches $c_i$, then $y$ will switch to being the responder on the following challenge). Thus $x \notin \mathrm{Opt}(G)$, and $A = \mathrm{Opt}(G)$ as required. $\qquad\square$

# B  Experimental details for Section 4

Here we provide an overview of the algorithms and games used for the experiments of Section 4. The algorithms used were the following (see Table 2).

- **pollackblair1998** [39]**:** A $(1 + 1)$-type CoEA with crossover. As our applications are not probabilistic, repeated play was not used in our implementation.
- **chellapillafogel1999** [6]**:** A $(\lambda + \lambda)$-type CoEA with players evaluated against random opponents. Self-adaptive features appearing in the original paper were not used in our implementation.
- **panaitluke2002** [37]**:** A population-based CoEA that uses a single-elimination tournament to evaluation individuals.
- **rosinbelew1997** [41]**:** A population-based CoEA with a unique evaluation method that promotes the retention of individuals that can defeat niches in the opposing population.
- **sjlk2015** [44]**:** A $(\lambda, 2\lambda)$-type CoEA which evaluates individuals based on average performance against the opposing population and "fitness sharing".
- **PDCoEA** [27]**:** A population-based CoEA which independently selects parents based on a game-theoretic pairwise dominance relation.
- **RankDivCoEA** [20]**:** A population-based CoEA which rewards individuals for inducing less common rankings of their opponents.

All non-EDA algorithms used the same mutation operator: given $x$, we obtain a mutant by, for each $u \in \mathrm{Int}(G)$, randomly resampling $x(u)$ uniformly from $F(u)$ with probability $\chi/n$, and otherwise leaving $x(u)$ unchanged. A constant mutation rate of $\chi = 0.3$ was used for all algorithms.

The games Subtraction Nim, Turning Turtles, Silver Dollar, and Chomp are all described in [4]. Tic-Tac-Toe is not often characterised as an *impartial* combinatorial game, owing to the fact that all possible game positions are reachable by only one player and the win condition being described positively rather than as the absence of a legal move. Nonetheless, it is easy to represent Tic-Tac-Toe as an ICG by creating an additional game position with no out-neighbours representing a lost game, and having the losing player sent there upon the completion of 3-in-a-row. As the second player can never force a win on a *strong positional game* (a class to which Tic-Tac-Toe belongs), draws were counted as a win for player two. A summary of relevant games is provided in Table 3.

Each experiment was conducted on an internal cluster provisioned with 1344 CPU cores and 6TB of RAM. The experiment used to produce the data presented in Figure 2 had a wall-clock time of 22 hours, resulting in a maximum 29568 core-hours. The experiment used to produce the data presented in Figure 3 had a wall-clock time of 8 hours, resulting in a maximum 10752 core-hours. Combined, the experiments required no more than 40320 core-hours.

# C  Supplementary material to Section 5

## C.1  An existing upper runtime bound for Tournament UMDA on RLO

The proof of the following runtime bound relies on the definitions of concepts including the *switchability* $s(v)$ of a vertex $v \in V$, the *depth* of a set of edges $A \subseteq E(G) := \{(u, v) : u \in V, v \in F(u)\}$,

Table 2: An overview of algorithms used in the experiments of Section 4.

| Label | Ref. | Applications | Paradigm | Archives | Crossover | Elitist |
|---|---|---|---|---|---|---|
| pollackblair1998 | [39] | Backgammon | one-pop. | ✓ | ✓ | |
| chellapillafogel1999 | [6] | Checkers, Chess | one-pop. | | | ✓ |
| panaitluke2002 | [37] | Nim | one-pop. | | ✓ | |
| rosinbelew1997 | [41] | Nim, 3D Tic-Tac-Toe | two-pop. | ✓ | ✓ | ✓ |
| sjlk2015 | [44] | Othello | two-pop. | | | |
| PDCoEA | [27] | | two-pop. | | | |
| RankDivCoEA | [20] | | two-pop. | | | |

Table 3: An overview of games used in the experiments of Section 4.

| Game | Notes | $\lvert V \rvert$ |
|---|---|---|
| Subtraction Nim | $n = 100, k = 2$ | 101 |
| RLO | 50 challenges | 201 |
| Turning Turtles | 6 coins | 64 |
| Silver Dollar | 9 squares, 4 coins | 126 |
| Chomp | $5 \times 5$ bar | 252 |
| Tic-Tac-Toe | | 629 |

and what it means for a set of edges to be a *v-switcher*. These notions are all defined in detail in Section 4 of [4].

**Proposition C.1.** *There is a constant $C > 0$ such that the following holds. Let $K > 0$ be an arbitrary constant, let $G$ be the game graph for* RLO*, and let $\mathcal{A}$ be described by Algorithm 1 where $c = 0$ and $\gamma = 1/(40n)$. Then, provided $\mu = C(K + 2)(40n)^3 \log n$,*

$$\mathbb{P}\left[T_{\mathcal{A}}^G(\mathrm{Opt}(G)) \leqslant C^2 40^5 (K + 2) n^5 \log^2 n\right] \geqslant 1 - n^{-K}.$$

*Proof.* The result is immediate from Corollary 5.3 of [4], provided we can verify that every vertex $v \in V$ has switchability $s(v)$ at most 1. However, this is indeed the case, as for each $v \in V$ the set $A_v := \{(u, v) : v \in F(u)\}$ is a $v$-switcher of depth 1. $\square$

## C.2 Empirical trajectories

Here we investigate how closely the actual trajectories of the vector

$$\mathbf{q}_t := (p_t(c_i, a_i), p_t(a_i, c_{i-1}), p_t(b_i, c_{i-1})) \in [\gamma, 1 - \gamma]^3$$

in a run of Tournament UMDA on RLO match the idealised deterministic version represented in Figure 4. Tournament UMDA was run with population size $\mu = 300$ on RLO with $n = 50$ challenges. This was repeated 15 times for both $c = 0$ and $c = 0.05$; the corresponding implied trajectories for $\mathbf{q}_t$ (for challenge number $i = 25$) are shown in the first two plots of Figure 6.

Both instances initialise with $\mathbf{q}_t$ in the middle of $[\gamma, 1 - \gamma]^3$, and the motion is somewhat akin to a random walk for a while. Once the algorithms have optimised the challenges up to $i - 1$, selective pressure on the entries of $\mathbf{q}_t$ kicks in and we see the trajectories of both algorithms follow curves similar to those in Figure 4. This is most clearly evident in the very direct movements seen at the edges of $[\gamma, 1 - \gamma]^3$ for the case $c = 0$. There are several notable differences in the behaviour between the two cases. The case $c = 0.05$ has $\mathbf{q}_t$ remain closer to $(0.5, 0.5, 0.5)$ before the application of selective pressure. Recalling the discussion of how setting $c > 0$ transforms UMDA from a balanced EDA to a stable one (see Section 2), this behaviour is similar to that explaining the speed-up of stable EDAs over balanced EDAs on the LEADINGONES benchmark [25]. After selective pressure is applied the $c = 0.05$ trajectory moves quickly and directly towards the set $\hat{A} := [\frac{1}{8}, \frac{7}{8}] \times [1 - 2\gamma, 1 - \gamma]^2$.

Perhaps most critically, once the $c = 0.05$ trajectory reaches $\hat{A}$ the bias towards uniformity (on the $p(c_i, a_i)$ axis) combined with the selective pressure (on the other two axes) causes the trajectory

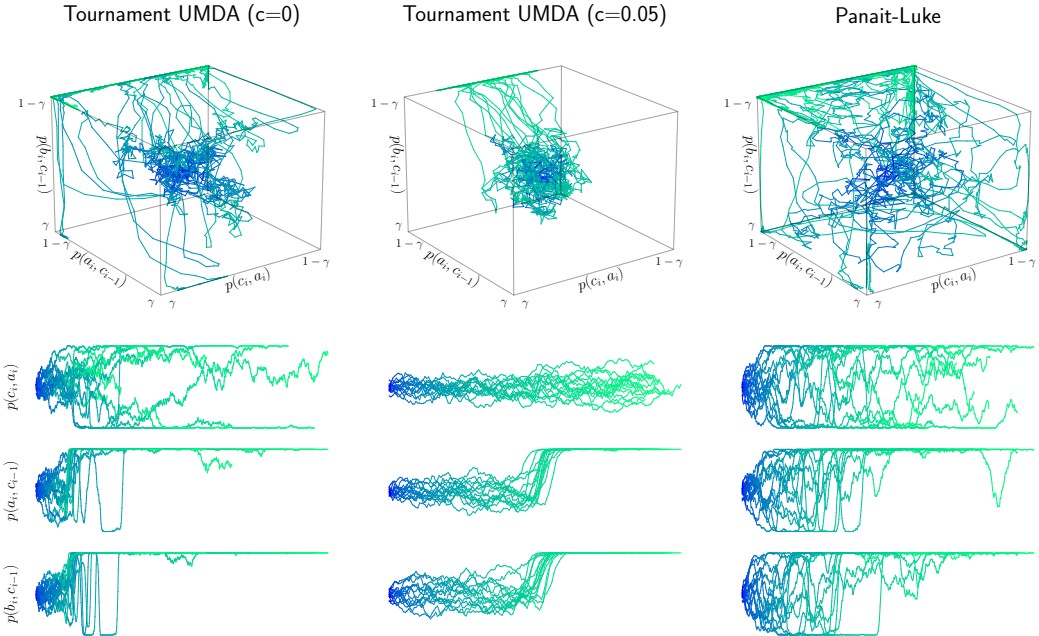

Figure 6: Observed trajectories of $\mathbf{q}_t$ for Tournament UMDA (in the cases $c = 0$ and $c = 0.05$) as well as the CoEA of [37]. Beneath the plots are projections of the trajectories onto each dimensions, with the horizontal axis representing time. We note that the horizontal axis has been scaled for each algorithm. Thus, while it may appear, for example, that the case $c = 0$ reaches the region $p(a_i, c_{i-1}) \approx p(b_i, c_{i-1})$ sooner, this is only relative to the overall runtime of the algorithm, which is in fact much longer.

to remain inside $\hat{A}$. This is in stark contrast to the case $c = 0$, where the $\mathbf{q}_t$ is not prevented from moving into the corners of the cube, from which point it is then even possible for $p(a_i, c_{i-1})$ and $p(b_i, c_{i-1})$ to drift significantly away from $1 - \gamma$, thus impeding further progress on the remaining challenges.

As a reference case, the corresponding trajectories for the CoEA of [37] (the best performing non-EDA from the experimental analysis of Section 4) is provided in the third plot of Figure 6. Here we observe an even more erratic motion in $[\gamma, 1 - \gamma]$, and the coevolutionary forgetting caused by $p(c_i, a_i)$ being too close to a boundary occurs more frequently.

## D   Proof of Theorem 5.2

### D.1   Further notation

Let us outline the notational conventions that will be adopted in the statements and proofs of this section. Throughout this section, $\mathcal{A}$ will always refer to Algorithm 1, and $G$ will always refer to the game graph for RLO. As per the statement of Theorem 5.2, we will always assume the following.

**A1**  $K > 0$ is an arbitrary constant.

**A2**  $c > 0$ is an arbitrary constant satisfying $c \leqslant 1/600$.

**A3**  $n \geqslant n_0$, where $n_0$ is a constant which is sufficiently large depending on $c$ and $K$.

**A4**  $\mu \geqslant \left( \frac{(K+4)10^5}{c^2} \right) n \log n$.

**A5**  $\gamma = 1/(100n)$.

It is helpful to define the following sets.

$$\mathcal{Q} := \prod_{v \in \text{Int}(G)} \mathcal{P}_\gamma(F(v)), \qquad \mathcal{Q}_0 := \prod_{v \in \text{Int}(G)} \mathcal{P}(F(v)).$$

Indeed, a run of $\mathcal{A}$ on RLO can now be considered as a stochastic process $(p_t)_{t=0}^\infty$ taking values in $\mathcal{Q}$ (with the intermediate frequencies $(q_t)_{t=1}^\infty$ arising in Algorithm 1 taking values in $\mathcal{Q}_0$). For notational convenience, given $p \in \mathcal{Q}$ we will usually write for $u, v \in V$,

$$p(u, v) = \begin{cases} p(u)(v) & \text{if } v \in F(u), \\ 0 & \text{otherwise.} \end{cases}$$

The process used to constrain $q_t \in \mathcal{Q}_0$ into $p_t \in \mathcal{Q}$ can be represented as a map $\overline{\pi}_\gamma : \mathcal{Q}_0 \to \mathcal{Q}$ defined as follows.

$$\overline{\pi}_\gamma(q)(u, v) = \pi_\gamma^{F(u)}(q(u))(v).$$

As this particular game graph always has $|F(u)| = 2$ whenever $u \in \text{Int}(G)$, we could equivalently have defined

$$\overline{\pi}_\gamma(q)(u, v) = \begin{cases} \min\{1 - \gamma, \max\{\gamma, q(u, v)\}\} & \text{if } v \in F(u), \\ 0 & \text{otherwise.} \end{cases}$$

It is therefore helpful to define the simple function $\hat{\pi} : [0, 1] \to [\gamma, 1 - \gamma]$ as

$$\hat{\pi}(x) = \min\{1 - \gamma, \max\{\gamma, x\}\}.$$

Given $p \in \mathcal{Q}$, we will use $w_p$ to represent the random variable taking values in $\mathcal{X}_G$ generated by sampling $x$ and $y$ independently according to $\text{Univ}(\mathcal{X}_G, p)$ and setting $w_p = x$ if $f_G(x, y) = 1$ and $w_p = y$ otherwise. Note that $w_{p_t}$ has the same distribution as a $w$ sampled in Algorithm 1.

Given a function $\eta : \mathcal{Q} \times V^2 \to \{-\infty\} \cup [-1, \frac{1}{2}]$, let $\phi_\eta : \mathcal{Q} \to \mathscr{P}(\mathcal{Q})$ be the map given by

$$\phi_\eta(p) = \left\{ \overline{\pi}_\gamma(q) : \begin{array}{l} q(u, v) \geqslant p(u, v) \cdot [1 + (\eta(p; u, v) + c(\frac{1}{5} - p(u, v))) \cdot (1 - p(u, v))] \\ \text{for every } u, v \in V \end{array} \right\}. \tag{1}$$

The motivation for this definition is that we will later establish that $p_{t+1} \in \phi_\eta(p_t)$ holds with high probability for carefully defined functions $\eta$. More precisely, we will see in Lemmas D.7 and D.9 that if $p_t = p$, then an individual $w$ sampled in Algorithm 1 satisfies equations of the form $\mathbb{P}(w(u) = v) = p(u, v) \cdot [1 + \eta(p; u, v) \cdot (1 - p(u, v))]$. After then accounting for the effect of $\text{MUTATE}(c, p_t, w)$ using Lemma D.3, we will then deduce that $\mathbb{E}[q_{t+1}(u, v)] \geqslant p(u, v) \cdot [1 + (\eta(p; u, v) + c(\frac{3}{8} - p(u, v))) \cdot (1 - p(u, v))]$. In this expression we relax $\frac{3}{8}$ to $\frac{1}{5}$ so that we can apply concentration inequalities in the proof of Lemma D.6. The only remaining feature to account for is then constraint steps of Algorithm 1, which motivates the inclusion of $\overline{\pi}_\gamma$ in the definition of $\phi_\eta$.

For some of the preliminary results (in particular, Lemmas D.4 and D.5) it is helpful to recursively define for $v \in V$,

$$f_G^v(x, y) = \begin{cases} -f_G^{x(v)}(y, x) & \text{if } v \in \text{Int}(G), \\ -1 & \text{otherwise,} \end{cases}$$

so that $f_G(x, y) = f_G^{c_n}(x, y)$. Intuitively, $f_G^v$ is the payoff function for the version of RLO where the initial position is $v$. It is also helpful to denote for $x, y \in \mathcal{X}_G$,

$$\text{Path}_G(x, y) = \{v_0, x(v_0), y(x(v_0)), x(y(x(v_0))), \ldots\}.$$

Finally, let $s_{1/2}^+ : \mathbb{R} \to \mathbb{R}$ be given by

$$s_{1/2}^+(x) = \begin{cases} x/2 & \text{if } x \geqslant 0, \\ x & \text{otherwise.} \end{cases}$$

Note that $s_{1/2}^+$ is a convenient tool for showing certain inequalities hold even when the signs of certain terms are unknown, and its role is not very important in the overall proof idea.

## D.2 Probabilistic tools

We quickly collect some probabilistic tools that will be useful later. The first is one of Bernstein's inequalities (see Theorem 6.6.1 of [47] for a more general version from which Theorem D.1 follows). The second is a simple bound on the variance of a binomial random variable.

**Theorem D.1.** *Let $C > 0$ and let $X_1, \ldots, X_n$ be independent random variables such that, for all $i$, $a_i \leqslant X_i \leqslant b_i$ almost surely and $b_i - a_i \leqslant C$. Set $S := \sum_{i \in [n]} X_i$ and $V := \mathrm{Var}[S] = \sum_{i \in [n]} \mathrm{Var}[X_i]$. Then, for any $t > 0$,*

$$\mathbb{P}(S \geqslant \mathbb{E}[S] + t) \leqslant \exp\left(-\frac{t^2/2}{V + C \cdot t/3}\right),$$

$$\mathbb{P}(S \leqslant \mathbb{E}[S] - t) \leqslant \exp\left(-\frac{t^2/2}{V + C \cdot t/3}\right).$$

**Proposition D.2.** *If $X \sim \mathrm{Bin}(\mu, p)$, then $\mathrm{Var}(X) \leqslant \min\{\mathbb{E}[X], \mu - \mathbb{E}[X]\}$.*

*Proof.* $\mathrm{Var}(X) = \mu p(1 - p) \leqslant \min\{\mu p, \mu(1 - p)\} = \min\{\mathbb{E}[X], \mu - \mathbb{E}[X]\}$. $\square$

## D.3 Preliminary lemmas for both bounds

The following lemma helps quantify the effect of the mutation operator in Algorithm 1.

**Lemma D.3.** *Suppose in Algorithm 1 that $0 \leqslant c \leqslant 1$, and $i \in I$ is such that $|S_i| = 2$. If $r \in S_i$, $a \leqslant 1/2$, and $p \in \prod_{i \in I} \mathcal{P}_\gamma(S_i)$ is such that*

$$\mathbb{P}(w(i) = r \mid p_t = p) \geqslant p(i, r) \cdot [1 + a \cdot (1 - p(i, r))],$$

*then*

$$\mathbb{E}[q_{t+1}(i, r) \mid p_t = p] \geqslant p(i, r) \cdot [1 + (a + c(\tfrac{3}{8} - p(i, r))) \cdot (1 - p(i, r))].$$

*Proof.* Let $q := p(i, r)$ be fixed and assume that $\mathbb{P}(w(i) = r \mid p_t = p) \geqslant q(1 + a(1 - q))$. Noting that $c \cdot \prod_{s \in S_i}(1 - p(i, s)) = cq(1 - q)$, We can compute

$$\begin{aligned}
\mathbb{E}[q_{t+1}(i, r) \mid p_t = p] &= \mathbb{P}(P_{t+1}(1)(i) = 1) = cq(1 - q) \cdot \tfrac{1}{2} + (1 - cq(1 - q)) \cdot \mathbb{P}(w(i) = 1) \\
&\geqslant cq(1 - q) \cdot \tfrac{1}{2} + (1 - cq(1 - q)) \cdot q(1 + a(1 - q)) \\
&= q \cdot [\tfrac{1}{2}c(1 - q) + 1 + a(1 - q) - cq(1 - q) - acq(1 - q)^2] \\
&= q \cdot [1 + (a + \tfrac{1}{2}c - cq - acq(1 - q)) \cdot (1 - q)] \\
&= q \cdot [1 + (a - acq(1 - q) + c(\tfrac{1}{2} - q)) \cdot (1 - q)] \\
&\geqslant q \cdot [1 + (a + c(\tfrac{3}{8} - q)) \cdot (1 - q)], \quad\quad\quad\quad\quad\quad\quad (2)
\end{aligned}$$

where in the final inequality we have used that $aq(1 - q) \leqslant 1/8$. $\square$

**Lemma D.4.** *Suppose that $p \in \mathcal{Q}$. Then, for any $u \in V$ and $v \in F(u)$,*

$$\mathbb{P}(w_p(u) = v) = p(u, v) \cdot [1 + \mathbb{P}(u \in \mathrm{Path}_G(x, y)) \cdot (1 - \mathbb{P}(f_G^v(x, y) = 1) - \mathbb{P}(f_G^u(x, y) = 1))]. \quad (3)$$

*Proof.* See Lemma 3.1 of [4]. $\square$

**Lemma D.5.** *Suppose that $p \in \mathcal{Q}$ and that $x$ and $y$ are sampled independently according to $\mathrm{Univ}(\mathcal{X}_G, p)$. Then for any $u \in V$ we have*

$$\mathbb{P}(f_G^u(x, y) = 1) = \sum_{v \in F(u)} p(u, v) \cdot (1 - \mathbb{P}(f_G^v(x, y) = 1)), \quad\quad (4)$$

$$= \sum_{v \in F(u)} \sum_{w \in F(v)} p(u, v) \cdot p(v, w) \cdot \mathbb{P}(f_G^w(x, y) = 1). \quad\quad (5)$$

*Proof.* (4) holds by the law of total probability. We can then deduce that

$$\sum_{v \in F(u)} p(u,v) \cdot (1 - \mathbb{P}(f_G^v(x,y) = 1)) = 1 - \sum_{v \in F(u)} p(u,v) \cdot \mathbb{P}(f_G^v(x,y) = 1)$$

$$\stackrel{(4)}{=} 1 - \sum_{v \in F(u)} p(u,v) \cdot \sum_{w \in F(v)} p(v,w) \cdot (1 - \mathbb{P}(f_G^w(x,y) = 1))$$

$$= \sum_{v \in F(u)} \sum_{w \in F(v)} p(u,v) \cdot p(v,w) \cdot \mathbb{P}(f_G^w(x,y) = 1),$$

and hence (5) also holds. $\qquad\square$

In the proof of the following lemma, we note that for real-valued random variables $X, Y$, we say that $X$ *stochastically dominates* $Y$, written $X \succcurlyeq Y$, if $\mathbb{P}(X \leqslant z) \leqslant \mathbb{P}(Y \leqslant z)$ holds for all $z \in \mathbb{R}$.

**Lemma D.6.** *Suppose that $\eta : \mathcal{Q} \times V^2 \to \{-\infty\} \cup [-1, \frac{1}{2}]$ is any function such that*

$$\mathbb{P}(w_p(u) = v) \geqslant p(u,v) \cdot [1 + \eta(p; u,v) \cdot (1 - p(u,v))] \tag{6}$$

*holds for every $p \in \mathcal{Q}$ and $(u,v) \in V^2$. Then, for every $p \in \mathcal{Q}$ we have $\mathbb{P}(p_{t+1} \notin \phi_\eta(p) \mid p_t = p) \leqslant n^{-K-2}$.*

*Proof.* Let $p$ be fixed. In the following proof, expectations and probabilities are all written conditional on the event that $p_t = p$. For every $(u,v) \in V^2$, the quantity $X_{u,v} := \mu \cdot q_{t+1}(u,v)$ is binomially distributed. Let us define

$$\varphi(u,v) = p(u,v) \cdot [1 + (\eta(p; u,v) + c(\tfrac{1}{5} - p(u,v))) \cdot (1 - p(u,v))]$$

For every $(u,v) \in V^2$, we will show that

$$\mathbb{P}\left(q_{t+1}(u,v) < \varphi(u,v)\right) \leqslant n^{-K-4}. \tag{7}$$

The lemma then follows using a union bound.

Noting that (7) is trivially true when $\eta(p; u,v) = -\infty$, we may additionally assume that $\eta(p; u,v) \in [-1, \frac{1}{2}]$. Using (6) together with Lemma D.3, we can establish that

$$\mathbb{E}[q_{t+1}(u,v)] \geqslant p(u,v) \cdot [1 + (\eta(p; u,v) + c(\tfrac{3}{8} - p(u,v))) \cdot (1 - p(u,v))]$$

$$\geqslant \varphi(u,v) + \frac{c}{6} \cdot p(u,v) \cdot (1 - p(u,v)). \tag{8}$$

For each $(u,v) \in V^2$, let

$$Y_{u,v} \sim \mathrm{Bin}(\,\mu\,,\ \varphi(u,v) + \tfrac{c}{6} \cdot p(u,v) \cdot (1 - p(u,v))\,).$$

$X_{u,v}$ and $Y_{u,v}$ are both binomial random variables with $\mu$ trials. In addition, (8) shows that $\mathbb{E}[X_{u,v}] \geqslant \mathbb{E}[Y_{u,v}]$. Therefore $X_{u,v} \succcurlyeq Y_{u,v}$, and hence

$$\mathbb{P}(q_{t+1}(u,v) < \varphi(u,v)) = \mathbb{P}(X_{u,v} < \mu \cdot \varphi(u,v)) \leqslant \mathbb{P}(Y_{u,v} < \mu \cdot \varphi(u,v))$$

$$\leqslant \mathbb{P}(Y_{u,v} < \mathbb{E}[Y_{u,v}] - \tfrac{c\mu}{6} \cdot p(u,v) \cdot (1 - p(u,v))).$$

In order to bound the final probability above, we need an upper bound on $\mathrm{Var}(Y_{u,v})$. For this, first observe that because $-1 \leqslant \eta(p; u,v) \leqslant 1/2$ we have

$$\mu \cdot p(u,v)^2 \stackrel{\textbf{A3}}{\leqslant} \mu \cdot \varphi(u,v) \leqslant \mathbb{E}[Y_{u,v}] \stackrel{\textbf{A3}}{\leqslant} 2\mu \cdot p(u,v). \tag{9}$$

Therefore,

$$\mathrm{Var}(Y_{u,v}) \stackrel{\text{Proposition D.2}}{\leqslant} \min\left\{\mathbb{E}[Y_{u,v}], \mu - \mathbb{E}[Y_{u,v}]\right\} \stackrel{(9)}{\leqslant} \min\left\{2\mu \cdot p(u,v), \mu \cdot (1 - p(u,v)^2)\right\}$$

$$= \min\left\{2\mu \cdot p(u,v), \mu \cdot (1 - p(u,v) + p(u,v) \cdot (1 - p(u,v)))\right\}$$

$$\leqslant \min\left\{2\mu \cdot p(u,v), 2\mu \cdot (1 - p(u,v))\right\} \leqslant 4\mu \cdot p(u,v) \cdot (1 - p(u,v)). \tag{10}$$

Therefore, applying Theorem D.1 (with $C = 1$ and $t = \frac{c\mu}{6} \cdot p(u, v) \cdot (1 - p(u, v)))$, we obtain

$$\mathbb{P}(Y_{u,v} < \mathbb{E}[Y_{u,v}] - \tfrac{c\mu}{6} \cdot p(u, v) \cdot (1 - p(u, v)))) \leqslant \exp\left(-\frac{t^2/2}{\mathrm{Var}(Y_{u,v}) + t/3}\right)$$

$$\leqslant \exp\left(-\frac{(\tfrac{c\mu}{6} \cdot p(u, v) \cdot (1 - p(u, v)))^2/2}{\mathrm{Var}(Y_{u,v}) + (\tfrac{c\mu}{6} \cdot p(u, v) \cdot (1 - p(u, v)))/3}\right)$$

$$\overset{(10)}{\leqslant} \exp\left(-\frac{(\tfrac{c\mu}{6} \cdot p(u, v) \cdot (1 - p(u, v)))^2/2}{(4 + \tfrac{c}{12})\mu \cdot p(u, v) \cdot (1 - p(u, v))}\right)$$

$$\leqslant \exp\left(-\frac{c^2\mu \cdot p(u, v) \cdot (1 - p(u, v))}{72(4 + \tfrac{c}{12})}\right) \leqslant \exp\left(-\frac{c^2\mu \cdot \gamma(1 - \gamma)}{72(4 + \tfrac{c}{12})}\right)$$

$$\overset{\mathbf{A5}}{=} \exp\left(-\frac{c^2\mu}{7200(4 + \tfrac{c}{12})(1 - \gamma)^{-1}n}\right) \overset{\mathbf{A2}}{\leqslant} \exp\left(-\frac{c^2\mu}{10^5 n}\right)$$

$$\overset{\mathbf{A4}}{\leqslant} \exp\left(-\frac{(K + 4)n \log n}{n}\right) = n^{-K-4}.$$

Hence (7) holds, thus completing the proof. $\qquad\square$

## D.4  Preliminary lemmas for the upper bound

A key tool in the proof of Theorem 5.2 will be a map $\Phi : \mathscr{P}(\mathcal{Q}) \to \mathscr{P}(\mathcal{Q})$, carefully defined so that for any set $A \subseteq \mathcal{Q}$, if $p_t \in A$ then $p_{t+1} \in \Phi(A)$ holds with high probability. The runtime result then follows by establishing that a repeated application of $\Phi$ eventually contracts $\mathcal{Q}$ down to a small set on which sampling an element of $\mathrm{Opt}(G)$ is very likely. This part of the argument is handled by the following two technical lemmas. Note that intuition behind the sets $(A_i)_{i=0}^n$ defined below (that is, that $A_i$ is representative of populations that are both diverse and strong in the final $i$ challenges) is provided in the discussion of Section 5.

**Lemma D.7.** *For* $0 \leqslant i \leqslant n$, *define*

$$A_i = \left\{ p \in \mathcal{Q} \ : \ \begin{array}{l} p(c_j, v) \geqslant \tfrac{1}{8} \text{ whenever } j \in [i] \text{ and } v \in \{a_j, b_j\}, \\ p(u, c_{j-1}) \geqslant 1 - \tfrac{1}{50n} \text{ whenever } j \in [i] \text{ and } u \in \{a_j, b_j\}. \end{array} \right\}$$

*Let* $\alpha : \mathcal{Q} \times V^2 \to \{-\infty\} \cup [-1, \tfrac{1}{2}]$ *be given by*

$$\alpha(p; u, v) = \begin{cases} s_{1/2}^+(p(b_i, c_{i-1}) - p(a_i, c_{i-1})) & \text{if } (u, v) = (c_i, a_i) \text{ and } p \in A_{i-1}, \\ s_{1/2}^+(p(a_i, c_{i-1}) - p(b_i, c_{i-1})) & \text{if } (u, v) = (c_i, b_i) \text{ and } p \in A_{i-1}, \\ \tfrac{1}{2}p(c_i, a_i) & \text{if } (u, v) = (a_i, c_{i-1}) \text{ and } p \in A_{i-1}, \\ \tfrac{1}{2}p(c_i, b_i) & \text{if } (u, v) = (b_i, c_{i-1}) \text{ and } p \in A_{i-1}, \\ -\infty & \text{otherwise.} \end{cases} \quad (11)$$

*Then, for any* $p \in \mathcal{Q}$ *and* $(u, v) \in V^2$ *we have*

$$\mathbb{P}(w_p(u) = v) \geqslant p(u, v) \cdot [1 + \alpha(p; u, v) \cdot (1 - p(u, v))].$$

*Proof of Lemma D.7.*  All cases where $\alpha(p; u, v) = -\infty$ are trivially true, which leaves us with four cases to verify. This will be done using Lemma D.4.

Given $u \in V$, let us write $N_u$ as a shorthand for the event $f_G^u(x, y) = 1$, where $x$ and $y$ are sampled independently according to $\mathrm{Univ}(\mathcal{X}_G, p)$. Note that if $u$ has an out-neighbourhood $F(u) = \{v, w\}$ of size 2, then with a view to applying Lemma D.4 we can observe that

$$1 - \mathbb{P}(N_v) - \mathbb{P}(N_u) \overset{(4)}{=} 1 - \mathbb{P}(N_v) - p(u, v) \cdot (1 - \mathbb{P}(N_v)) - (1 - p(u, v)) \cdot (1 - \mathbb{P}(N_w))$$
$$= (1 - p(u, v)) \cdot (\mathbb{P}(N_w) - \mathbb{P}(N_v)). \quad (12)$$

Let $p \in A_{i-1}$. Before verifying the four critical cases, let us collect some useful identities. In order for $x$ to win when beginning the game from $c_{i-1}$, $y$ must make at least one mistake as the responder

in the final $i-1$ challenges. Because $p \in A_{i-1}$, it holds for every $j \in [i-1]$ and $u \in \{a_j, b_j\}$ that $\mathbb{P}(y(u) \neq c_{j-1}) \leqslant \frac{1}{50n}$. Therefore, we have

$$\mathbb{P}(N_{c_{i-1}}) \leqslant \mathbb{P}(y(u) \neq c_{j-1} \text{ for some } j \in [i-1] \text{ and } u \in \{a_j, b_j\}) \leqslant \frac{2(i-1)}{50n} \leqslant \frac{1}{10}, \quad (13)$$

and

$$\mathbb{P}(N_{d_i}) \stackrel{(4)}{=} 1 - \mathbb{P}(N_{c_{i-1}}). \quad (14)$$

We can now compute

$$\begin{aligned}
1 - \mathbb{P}(N_{c_{i-1}}) - \mathbb{P}(N_{a_i}) &\stackrel{(12)}{=} (1 - p(a_i, c_{i-1})) \cdot (\mathbb{P}(N_{d_i}) - \mathbb{P}(N_{c_{i-1}})) \\
&\stackrel{(14)}{=} (1 - p(a_i, c_{i-1})) \cdot (1 - 2 \cdot \mathbb{P}(N_{c_{i-1}})),
\end{aligned} \quad (15)$$

and similarly,

$$1 - \mathbb{P}(N_{c_{i-1}}) - \mathbb{P}(N_{b_i}) \stackrel{(12),(14)}{=} (1 - p(b_i, c_{i-1})) \cdot (1 - 2 \cdot \mathbb{P}(N_{c_{i-1}})). \quad (16)$$

Next,

$$\begin{aligned}
\mathbb{P}(N_{b_i}) - \mathbb{P}(N_{a_i}) &= [1 - \mathbb{P}(N_{c_{i-1}}) - \mathbb{P}(N_{a_i})] - [1 - \mathbb{P}(N_{c_{i-1}}) - \mathbb{P}(N_{b_i})] \\
&\stackrel{(15),(16)}{=} (1 - p(a_i, c_{i-1}) \cdot (1 - 2 \cdot \mathbb{P}(N_{c_{i-1}})) - (1 - p(b_i, c_{i-1}) \cdot (1 - 2 \cdot \mathbb{P}(N_{c_{i-1}})) \\
&= (1 - 2 \cdot \mathbb{P}(N_{c_{i-1}})) \cdot (p(b_i, c_{i-1}) - p(a_i, c_{i-1})).
\end{aligned} \quad (17)$$

The cases are then verified by applying Lemma D.4 as follows. If $(u, v) = (c_i, a_i)$, then

$$\begin{aligned}
\mathbb{P}(w_p(c_i) = a_i) &\stackrel{(3)}{=} p(c_i, a_i) \cdot [1 + (1 - \mathbb{P}(N_{a_i}) - \mathbb{P}(N_{c_i}))] \\
&\stackrel{(12)}{=} p(c_i, a_i) \cdot [1 + (\mathbb{P}(N_{b_i}) - \mathbb{P}(N_{a_i})) \cdot (1 - p(c_i, a_i))] \\
&\stackrel{(17)}{=} p(c_i, a_i) \cdot [1 + (1 - 2 \cdot \mathbb{P}(N_{c_{i-1}})) \cdot (p(b_i, c_{i-1}) - p(a_i, c_{i-1})) \cdot (1 - p(c_i, a_i))] \\
&\stackrel{(13)}{\geqslant} p(c_i, a_i) \cdot [1 + \alpha(p; c_i, a_i) \cdot (1 - p(c_i, a_i))].
\end{aligned}$$

If $(u, v) = (c_i, b_i)$, a similar computation to the above also yields $\mathbb{P}(w_p(c_i) = b_i) \geqslant p(c_i, b_i) \cdot [1 + \alpha(p; c_i, b_i) \cdot (1 - p(c_i, b_i))]$. If $(u, v) = (a_i, c_{i-1})$, then

$$\begin{aligned}
\mathbb{P}(w_p(a_i) = c_{i-1}) &\stackrel{(3)}{=} p(a_i, c_{i-1}) \cdot [1 + \mathbb{P}(a_i \in \text{Path}_G(x, y)) \cdot (1 - \mathbb{P}(N_{c_{i-1}}) - \mathbb{P}(N_{a_i})] \\
&\stackrel{(15)}{=} p(a_i, c_{i-1}) \cdot [1 + p(c_i, a_i) \cdot (1 - 2 \cdot \mathbb{P}(N_{c_{i-1}})) \cdot (1 - p(a_i, c_{i-1}))] \\
&\stackrel{(13)}{\geqslant} p(a_i, c_{i-1}) \cdot [1 + \alpha(p; a_i, c_{i-1}) \cdot (1 - p(a_i, c_{i-1}))].
\end{aligned}$$

Finally, if $(u, v) = (b_i, c_{i-1})$, then

$$\begin{aligned}
\mathbb{P}(w_p(b_i) = c_{i-1}) &\stackrel{(3)}{=} p(b_i, c_{i-1}) \cdot [1 + \mathbb{P}(b_i \in \text{Path}_G(x, y)) \cdot (1 - \mathbb{P}(N_{c_{i-1}}) - \mathbb{P}(N_{b_i})] \\
&\stackrel{(16)}{=} p(b_i, c_{i-1}) \cdot [1 + p(c_i, b_i) \cdot (1 - 2 \cdot \mathbb{P}(N_{c_{i-1}})) \cdot (1 - p(b_i, c_{i-1}))] \\
&\stackrel{(13)}{\geqslant} p(b_i, c_{i-1}) \cdot [1 + \alpha(p; b_i, c_{i-1}) \cdot (1 - p(b_i, c_{i-1}))].
\end{aligned}$$

$\square$

**Lemma D.8.** *Let the set $A_n$ and the function $\alpha : \mathcal{Q} \times V^2 \to \{-\infty\} \cup [-1, \frac{1}{2}]$ be defined as in Lemma D.7. Let $\Phi : \mathscr{P}(\mathcal{Q}) \to \mathscr{P}(\mathcal{Q})$ be the map given by $\Phi(A) = \cup_{p \in A} \phi_\alpha(p)$. Fix $\tau = \frac{1}{100} \log^2 (1/\gamma)$.*

*It then holds that $\Phi^{8\tau n}(\mathcal{Q}) \subseteq A_n$.*

The proof of Lemma D.8 is by far the most detailed part of this section. Additionally, because it follows only from the definition of $\Phi$, it does not necessarily provide direct intuition for the dynamics of Algorithm 1 and its application to Reciprocal LeadingOnes. For these reasons, we opt to first present the overall proof of Theorem 5.2, before proving Lemma D.8 later in Section D.7.

### D.5 Preliminary lemmas for the lower bound

Whereas the sets $(A_i)_{i=0}^n$ are representative of populations that are both diverse and strong in *at least* the final $i$ challenges, the sets $(B_i)_{i=0}^n$ in the following lemma are representative of populations which are strong in *at most* the final $i$ challenges.

**Lemma D.9.** *Let $\ell \geqslant 4$. For $0 \leqslant i \leqslant n$, define*

$$B_i = \{p \in \mathcal{Q} : p(u,v) \geqslant \tfrac{1}{8} \text{ whenever } j > i,\ u \in \{a_j, b_j\},\ \text{and } v \in \{d_j, c_{j-1}\}\}.$$

*Let $\beta : \mathcal{Q} \times V^2 \to \{-\infty\} \cup [-1, \tfrac{1}{2}]$ be given by*

$$\beta(p; u, v) = \begin{cases} -2(3/4)^{\ell-1} & \text{if } u \in \{a_j, b_j\},\ v \in \{d_j, c_{j-1}\},\ \text{and } p \in B_{j-\ell}, \\ -\infty & \text{otherwise.} \end{cases} \tag{18}$$

*Then, for any $p \in \mathcal{Q}$ and $(u,v) \in V^2$ we have*

$$\mathbb{P}(w_p(u) = v) \geqslant p(u,v) \cdot [1 + \beta(p; u, v) \cdot (1 - p(u,v))].$$

*Proof.* As with the proof of Lemma D.7, given $u \in V$ we will use $N_u$ to denote the event $f_G^u(x,y) = 1$. Note that the identities (12) and (14) apply here also.

We require the following claim.

**Claim D.10.** *For $0 \leqslant j \leqslant n$ and $0 \leqslant k \leqslant j$, if $p \in B_{j-k}$ and $v \in \{c_j, d_{j+1}\}$, then $|\mathbb{P}(N_v) - \tfrac{1}{2}| \leqslant (3/4)^k$.*

*Proof of Claim D.10.* By (14), it suffices to prove the claim holds for $v = c_j$. We prove this by induction on $k$, noting that the case $k = 0$ is trivial. For the inductive case, assume for $k > 0$ that $p \in B_{j-k}$ and $|\mathbb{P}(N_{c_{j-1}}) - \tfrac{1}{2}| \leqslant (3/4)^{k-1}$. Writing $\delta = \tfrac{1}{2} - \mathbb{P}(N_{c_{j-1}})$, we can now compute

$$\mathbb{P}(N_{c_j}) \overset{(4)}{=} \sum_{v \in \{a_j, b_j\}} \sum_{w \in \{d_j, c_{j-1}\}} p(c_j, v) \cdot p(v, w) \cdot \mathbb{P}(N_w)$$

$$= \sum_{v \in \{a_j, b_j\}} p(c_j, v) \cdot \big(p(v, d_j) \cdot \mathbb{P}(N_{d_j}) + p(v, c_{j-1}) \cdot \mathbb{P}(N_{c_{j-1}})\big)$$

$$\overset{(14)}{=} \sum_{v \in \{a_j, b_j\}} p(c_j, v) \cdot \big((1 - p(v, c_{j-1})) \cdot (1 - \mathbb{P}(N_{c_{j-1}})) + p(v, c_{j-1}) \cdot \mathbb{P}(N_{c_{j-1}})\big)$$

$$= \sum_{v \in \{a_j, b_j\}} p(c_j, v) \cdot \big((1 - p(v, c_{j-1})) \cdot (\tfrac{1}{2} + \delta) + p(v, c_{j-1}) \cdot (\tfrac{1}{2} - \delta)\big)$$

$$= \sum_{v \in \{a_j, b_j\}} p(c_j, v) \cdot \big(\tfrac{1}{2} + (1 - 2p(v, c_{j-1})) \cdot \delta\big)$$

$$= \frac{1}{2} + \sum_{v \in \{a_j, b_j\}} p(c_j, v) \cdot (1 - 2p(v, c_{j-1})) \cdot \delta. \tag{19}$$

Because $p \in B_{j-k} \subseteq B_{j-1}$ it holds for each $v \in \{a_j, b_j\}$ that $p(v, c_{j-1}) \in [\tfrac{1}{8}, \tfrac{7}{8}]$, and hence $|1 - 2p(v, c_{j-1})| \leqslant 3/4$. Therefore,

$$|\mathbb{P}(N_{c_j}) - \tfrac{1}{2}| \overset{(19)}{\leqslant} \sum_{v \in \{a_j, b_j\}} p(c_j, v) \cdot |1 - 2p(v, c_{j-1})| \cdot |\delta| \leqslant \tfrac{3}{4}|\delta| \leqslant (3/4)^k,$$

as required. $\quad\boxdot$

We now turn to proving the lemma. All cases where $\beta(p; u, v) = -\infty$ are trivially true, and so we only need to consider the case where $u \in \{a_j, b_j\}$, $v \in \{d_j, c_{j-1}\}$, and $p \in B_{j-\ell}$. Under these assumptions, we have

$$\mathbb{P}(w_p(u) = d_j) \overset{(3)}{=} p(u, d_j) \cdot [1 + (1 - \mathbb{P}(N_{d_j}) - \mathbb{P}(N_u))]$$

$$\overset{(12)}{=} p(u, d_j) \cdot [1 + (\mathbb{P}(N_{c_{j-1}}) - \mathbb{P}(N_{d_j})) \cdot (1 - p(u, d_j))]$$

$$\overset{\text{Claim D.10}}{\geqslant} p(u, d_j) \cdot [1 - 2(3/4)^{\ell-1} \cdot (1 - p(u, d_j))].$$

By swapping $d_j$ and $c_{j-1}$ in the above calculation, we also obtain $\mathbb{P}(w_p(u) = c_{j-1}) \geqslant p(u, c_{j-1}) \cdot [1 - 2(3/4)^{\ell-1} \cdot (1 - p(u, c_{j-1}))].$ $\qquad\square$

Recall in the following lemma that the definition of $\phi_\beta$ is provided by (1).

**Lemma D.11.** *Let* $\ell = \lceil 4 \log{(200/c)} \rceil$. *Let the sets* $(B_i)_{i=0}^n$ *and the function* $\beta : \mathcal{Q} \times V^2 \to \{-\infty\} \cup [-1, \frac{1}{2}]$ *be defined as in Lemma D.9. Let* $\Psi : \mathscr{P}(\mathcal{Q}) \to \mathscr{P}(\mathcal{Q})$ *be the map given by* $\Psi(A) = \cup_{p \in A} \phi_\beta(p)$.

*For every* $0 \leqslant i \leqslant n - \ell$, *it holds that* $\Psi(B_i) \subseteq B_{i+\ell}$.

*Proof.* First, note that

$$\ell \geqslant 4 \log{(200/c)} \overset{\mathbf{A2}}{\geqslant} \frac{\log{(200/c)}}{\log{(4/3)}} + 1 = \frac{\log{(c/200)}}{\log{(3/4)}} + 1 \tag{20}$$

Suppose that $q \in \Psi(B_i)$, so that there exists $p \in B_i$ such that $q \in \phi_\beta(p)$. In particular, there is some $q' \in \mathcal{Q}_0$ such that

$$q'(u, v) \geqslant p(u, v) \cdot [1 + (\beta(p; u, v) + c(\tfrac{1}{5} - p(u, v))) \cdot (1 - p(u, v))] \quad \text{for every } u, v \in V, \tag{21}$$
$$\bar{\pi}_\gamma(q') = q. \tag{22}$$

We will show that $q \in B_{i+\ell}$. To do this we need to confirm that $q(u, v) \geqslant \frac{1}{8}$ whenever $j > i + \ell$, $u \in \{a_j, b_j\}$, and $\{d_j, c_{j-1}\}$. So let $j > i + \ell$, $u \in \{a_j, b_j\}$, and $v \in \{d_j, c_{j-1}\}$ be fixed, and observe that because $p \in B_i \subseteq B_{j-\ell}$,

$$\beta(p; u, v) \overset{(18)}{=} -2(3/4)^{\ell-1} \overset{(20)}{\geqslant} -\frac{c}{100}. \tag{23}$$

Therefore,

$$q(u, v) \overset{(22)}{=} \hat{\pi}(q'(u, v)) \overset{(21)}{\geqslant} \hat{\pi}(p(u, v) \cdot [1 + (\beta(p; u, v) + c(\tfrac{1}{5} - p(u, v))) \cdot (1 - p(u, v))])$$
$$\overset{(23)}{\geqslant} \hat{\pi}(p(u, v) \cdot [1 + c(\tfrac{3}{16} - p(u, v)) \cdot (1 - p(u, v))]). \tag{24}$$

There are now two cases to consider. First, if $p(u, v) \geqslant 3/16$ we have

$$q(u, v) \overset{(24)}{\geqslant} \hat{\pi}(p(u, v) \cdot [1 - c]) \geqslant \hat{\pi}(\tfrac{3}{16} \cdot [1 - c]) \geqslant \hat{\pi}(1/8) = 1/8.$$

On the other hand, if $1/8 \leqslant p(u, v) \leqslant 3/16$ then

$$q(u, v) \overset{(24)}{\geqslant} \hat{\pi}(p(u, v)) \geqslant \hat{\pi}(1/8) = 1/8.$$

In either case, we have $q(u, v) \geqslant \frac{1}{8}$ whenever $j > i + \ell$. Thus $q \in B_{i+\ell}$ for every $q \in \Psi(B_i)$, and hence $\Psi(B_i) \subseteq B_{i+\ell}$, as required.

$\qquad\square$

The following lemma shows that if $p_t \in B_{n/4}$ describes the current state of Algorithm 1, then the probability that an optimal strategy is found in the following generation is very small.

**Lemma D.12.** *If* $p \in B_{n/2}$ *and* $\mu \leqslant n^3$, *then* $\mathbb{P}(P_{t+1} \cap \mathrm{Opt}(G) \neq \emptyset \mid p_t = p) \leqslant n^{-K-2}$.

*Proof.* Let us assume that $p_t = p$ for some $p \in B_{n/4}$. Let $j \in [\mu]$ be fixed. We need to estimate $\mathbb{P}(P_{t+1}(j) \in \mathrm{Opt}(G))$, where we recall from Algorithm 1 that $P_{t+1}(j) \sim \text{MUTATE}(c, p, w_p)$. As in Algorithm 1, let $x$ and $y$ both be sampled independently according to $\text{Univ}(\mathcal{X}, p)$, and let us assume that $x' \sim \text{MUTATE}(c, p, x)$ and $y' \sim \text{MUTATE}(c, p, y)$. Note that with this notation, we have $P_{t+1}(j) \in \{x', y'\}$.

Repeating the calculations of Lemma D.3 (with $a = 0$), we obtain for each $i \geqslant n/2$,

$$\mathbb{P}(x'(a_i) = d_i) \geqslant p(a_i, d_i) \cdot [1 + c(\tfrac{1}{4} - p(a_i, d_i)) \cdot (1 - p(a_i, d_i))] \geqslant \tfrac{1}{8}, \tag{25}$$

where we have used the fact that $p(a_i, d_i) \geqslant 1/8$ (as $p \in B_{n/2}$). The value of $x'(u)$ is independent for every $u \in V$, and hence using the characterisation of $\mathrm{Opt}(G)$ provided by Proposition 3.2,

$$\mathbb{P}(x' \in \mathrm{Opt}(G)) \leqslant \mathbb{P}\left(\bigwedge_{i \in n} x'(a_i) = c_{i-1}\right) \leqslant \prod_{i > n/2} \mathbb{P}(x'(a) = c_{i-1})$$

$$\leqslant \prod_{i > n/2} (1 - \mathbb{P}(x'(a) = d_i)) \overset{(25)}{\leqslant} (7/8)^{n/2} \overset{\mathbf{A3}}{\leqslant} \tfrac{1}{2}n^{-K-5}.$$

Similarly, we also have $\mathbb{P}(y' \in \mathrm{Opt}(G)) \leqslant \frac{1}{2}n^{-K-4}$, and hence

$$\mathbb{P}(P_{t+1}(j) \in \mathrm{Opt}(G)) \leqslant \mathbb{P}(x' \in \mathrm{Opt}(G)) + \mathbb{P}(y' \in \mathrm{Opt}(G)) \leqslant n^{-K-5}.$$

Taking a union bound over all $j \in [\mu]$ then yields the desired result. $\qquad\square$

### D.6 Proof of Theorem 5.2

*Proof of Theorem 5.2.* First, we consider the upper bound $\mathbb{P}[T_{\mathcal{A}}^G(\mathrm{Opt}(G)) > \mu n(\log n)^2]$. Let the sets $(A_i)_{i=0}^n$ and the function $\alpha : \mathcal{Q} \times V^2 \to \{-\infty\} \cup [-1, \frac{1}{2}]$ be defined as in Lemma D.7, and note that we have $A_n \subseteq \ldots \subseteq A_0 = \mathcal{Q}$. If for some $t$ we have $p_t \in A_n$, then $1 - \frac{1}{50n} \leqslant p_t(u, c_{i-1}) \leqslant q_t(u, c_{i-1})$ whenever $u \in \{a_i, b_i\}$ for $i \in [n]$, and hence

$$\tfrac{1}{\mu} \sum_{j \in [\mu]} \sum_{i \in [n]} \sum_{u \in \{a_i, b_i\}} \mathbb{1}(P_t(j)(u) = c_{i-1}) = \sum_{i \in [n]} \sum_{u \in \{a_i, b_i\}} q_t(u, c_{i-1})$$

$$\geqslant 2n(1 - \tfrac{1}{50n}) > 2n - 1.$$

It would then follow that there is some $j$ such that $\sum_{i \in [n]} \sum_{u \in \{a_i, b_i\}} \mathbb{1}(P_t(j)(u) = c_{i-1}) = 2n$, and hence $P_t(j) \in A_{\mathrm{opt}}$. Therefore

$$T_{\mathcal{A}}^G(\mathrm{Opt}(G)) \leqslant \mu \cdot \min\{t : p_t \in A_n\}. \tag{26}$$

By Lemma D.7, we have that $\mathbb{P}(w_p(u) = v) \geqslant p(u, v) \cdot [1 + \alpha(p; u, v) \cdot (1 - p(u, v))]$ for every $p \in \mathcal{Q}$ and $(u, v) \in V^2$. In particular, the condition for Lemma D.6 is satisfied for $\eta = \alpha$, and so it holds for every $p \in \mathcal{Q}$ that

$$\mathbb{P}(p_{t+1} \notin \phi_\alpha(p) \mid p_t = p) \leqslant n^{-K-2}. \tag{27}$$

Let $\tau = \frac{(\log(1/\gamma))^2}{100}$, so that

$$8\tau n = \frac{8n(\log(100n))^2}{100} \leqslant n(\log n)^2. \tag{28}$$

Let $\Phi : \mathscr{P}(\mathcal{Q}) \to \mathscr{P}(\mathcal{Q})$ be the map given by $\Phi(A) = \cup_{p \in A} \phi_\alpha(p)$. By Lemma D.8, $\Phi^{8\tau n}(\mathcal{Q}) \subseteq A_n$. Consequently, if $p_{8\tau n} \notin A_n$, then there must exist some $t < 8\tau n$ such that $p_{t+1} \notin \phi_\alpha(p_t)$. Therefore, we have

$$\mathbb{P}[T_{\mathcal{A}}^G(\mathrm{Opt}(G)) > \mu n(\log n)^2] \overset{(26)}{\leqslant} \mathbb{P}[\mu \cdot \min\{t : p_t \in A_n\} > \mu \cdot n(\log n)^2]$$

$$\overset{(28)}{=} \mathbb{P}[\min\{t : p_t \in A_n\} > 8\tau n] \leqslant \mathbb{P}(p_{8\tau n} \notin A_n)$$

$$\leqslant \mathbb{P}\left(\bigvee_{t < 8\tau n}(p_{t+1} \notin \phi_\alpha(p_t))\right)$$

$$\overset{(27)}{\leqslant} n^{-K-2} \cdot 8n\tau \leqslant \tfrac{1}{2}n^{-K}.$$

Next we consider the lower bound $\mathbb{P}[T_{\mathcal{A}}^G(\mathrm{Opt}(G)) < n^2 \log n]$. Note that

$$n^2 \log n \overset{\mathbf{A2}}{\leqslant} \frac{\left(\frac{10^5}{c^2}\right) n^2 \log n}{2 \cdot \lceil 4 \log(200/c) \rceil} \overset{\mathbf{A1,A4}}{\leqslant} \frac{\mu n}{2 \cdot \lceil 4 \log(200/c) \rceil} \tag{29}$$

If $\mu > n^3$ then by definition we have $T_{\mathcal{A}}^G(\mathrm{Opt}(G)) \geqslant n^3$, and so we may additionally assume that $\mu \leqslant n^3$. Let the sets $(B_i)_{i=1}^n$ and the function $\beta : \mathcal{Q} \times V^2 \to \{-\infty\} \cup [-1, \frac{1}{2}]$ be defined as in

Lemma D.9, and note that we have $B_0 \subseteq \ldots \subseteq B_n = \mathcal{Q}$. Deterministically, it holds that $p_0 \in B_n$. By Lemma D.9, we have that $\mathbb{P}(w_p(u) = v) \geqslant p(u, v) \cdot [1 + \beta(p; u, v) \cdot (1 - p(u, v))]$ for every $p \in \mathcal{Q}$ and $(u, v) \in V^2$. In particular, the condition for Lemma D.6 is satisfied for $\eta = \beta$, and so it holds for every $p \in \mathcal{Q}$ that

$$\mathbb{P}(p_{t+1} \notin \phi_\beta(p) \mid p_t = p) \leqslant n^{-K-2}. \tag{30}$$

Let $\Psi : \mathscr{P}(\mathcal{Q}) \to \mathscr{P}(\mathcal{Q})$ be the map given by $\Psi(A) = \cup_{p \in A} \phi_\beta(p)$. Suppose that $0 \leqslant t \leqslant n/(2\ell)$, where we define $\ell = \lceil 4 \log (200/c) \rceil$. By Lemma D.11, $\Psi^t(\mathcal{Q}) \subseteq B_{\ell t}$. Consequently, if $p_t \notin B_{\ell t}$, then there must exist some $s < t$ such that $p_{s+1} \notin \phi_\beta(p_s)$. Therefore, we have

$$\mathbb{P}(p_t \notin B_{\ell t}) \leqslant \mathbb{P}\left( \bigvee_{s<t} (p_{s+1} \notin \phi_\beta(p_s)) \right) \overset{(30)}{\leqslant} n^{-K-2} \cdot t \leqslant \tfrac{1}{4} n^{-K-1}. \tag{31}$$

Next, note that if $p \in B_{\ell t}$, then as $B_{\ell t} \subseteq B_{3n/4}$ and $\mu \leqslant n^3$ we can apply Lemma D.12 to obtain $\mathbb{P}(P_{t+1} \cap \mathrm{Opt}(G) \neq \emptyset \mid p_t = p) \leqslant n^{-K-2}$. Therefore, we may deduce for any $t < n/(2\ell)$ that

$$
\begin{aligned}
\mathbb{P}(P_{t+1} \cap \mathrm{Opt}(G) \neq \emptyset) &= \mathbb{P}(P_{t+1} \cap \mathrm{Opt}(G) \neq \emptyset \mid p_t \in B_{\ell t}) \cdot \mathbb{P}(p_t \in B_{\ell t}) + \\
&\quad \mathbb{P}(P_{t+1} \cap \mathrm{Opt}(G) \neq \emptyset \mid p_t \notin B_{\ell t}) \cdot \mathbb{P}(p_t \notin B_{\ell t}) \\
&\leqslant \mathbb{P}(P_{t+1} \cap \mathrm{Opt}(G) \neq \emptyset \mid p_t \in B_{\ell t}) + \mathbb{P}(p_t \notin B_{\ell t}) \\
&\overset{(31)}{\leqslant} n^{-K-2} + \tfrac{1}{4} n^{-K-1} \leqslant \tfrac{1}{2} n^{-K-1}.
\end{aligned}
$$

Therefore, we may compute

$$
\begin{aligned}
\mathbb{P}[T_{\mathcal{A}}^G(\mathrm{Opt}(G)) < n^2 \log n] &\overset{(29)}{\leqslant} \mathbb{P}[T_{\mathcal{A}}^G(\mathrm{Opt}(G)) \leqslant \mu n/(2\ell)] \\
&= \mathbb{P}\left( \bigvee_{t<n/(2\ell)} (P_{t+1} \cap \mathrm{Opt}(G) \neq \emptyset) \right) \\
&\leqslant \sum_{t<n/(2\ell)} \mathbb{P}(P_{t+1} \cap \mathrm{Opt}(G) \neq \emptyset) \leqslant \frac{n}{2\ell} \cdot \frac{1}{2} n^{-K-1} \leqslant \frac{1}{2} n^{-K}.
\end{aligned}
$$

The theorem now follows by combining the upper and lower bounds using a union bound. $\square$

### D.7 Proof of Lemma D.8

From here, we will use $\phi$ as a shorthand for $\phi_\alpha$.

It will be useful to map probabilities in the interval $[\gamma, 1 - \gamma]$ onto logit space using the function $g$ defined in the following lemma. Here we collect some simple properties of this mapping.

**Lemma D.13.** *Let $0 < \gamma < 1/2$ be fixed, and let $g : [\gamma, 1 - \gamma] \to \mathbb{R}$ be defined by*

$$g(y) = \log \left( \frac{y}{1 - y} \right).$$

*Then the following properties hold.*

**B1** *$g$ is strictly increasing*

**B2** *$|g(y)| \leqslant \log (1/\gamma)$ for any $y \in [\gamma, 1 - \gamma]$.*

**B3** *For any $a \in [0, 1]$ and $y \in [\gamma, 1 - \gamma]$, we have $g(y(1 + a(1 - y))) - g(y) \geqslant a/2$.*

**B4** *For any $a \in [0, 1]$ and $y \in [\gamma, 1 - \gamma]$, we have $g(y(1 - a(1 - y))) - g(y) \leqslant -a/2$.*

**B5** *For any $a \in [0, \frac{1}{2}]$ and $y \in [\gamma, 1 - \gamma]$, we have $g(y(1 - a(1 - y))) - g(y) \geqslant -2a$.*

**B6** *If $\gamma \leqslant y \leqslant \frac{1}{2}$, then $-1 - \log y \leqslant g(1 - y) \leqslant -\log y$.*

*Proof.* Because $g(y) = \log \left( \frac{y}{1-y} \right) = \log y - \log (1 - y)$, **B1** follows from the fact that $\log$ is a strictly increasing function. Using this, we can deduce that for any $y \in [\gamma, 1 - \gamma]$,

$$|g(y)| \leqslant \max \{|g(\gamma)|, |g(1 - \gamma)|\} = \max \{|\log (\tfrac{\gamma}{1-\gamma})|, |\log (\tfrac{1-\gamma}{\gamma})|\} = \log (\tfrac{1-\gamma}{\gamma}) \leqslant \log (1/\gamma),$$

and so **B2** holds. Next, observe that for any $a \in [-1, 1]$ and $y \in [\gamma, 1 - \gamma]$,

$$
\begin{aligned}
g(y(1 + a(1 - y))) - g(y) &= \log\left(\frac{y(1 + a(1 - y))}{1 - y(1 + a(1 - y))}\right) - \log\left(\frac{y}{1 - y}\right) \\
&= \log\left(\frac{(1 - y)(1 + a(1 - y))}{1 - y(1 + a(1 - y))}\right) = \log\left(\frac{(1 - y)(1 + a - ay)}{1 - (a + 1)y + ay^2}\right) \\
&= \log\left(\frac{(1 - y)(1 + a - ay)}{(1 - y)(1 - ay)}\right) = \log\left(\frac{1 + a - ay}{1 - ay}\right) \\
&= \log\left(1 + \frac{a}{1 - ay}\right).
\end{aligned}
\tag{32}
$$

If $a \in [0, 1]$ and $y \in [\gamma, 1 - \gamma]$ then

$$
g(y(1 + a(1 - y))) - g(y) \overset{(32)}{=} \log\left(1 + \frac{a}{1 - ay}\right) \geqslant \log(1 + a) \geqslant a/2,
$$

and so **B3** holds. If $a \in [0, 1]$ and $y \in [\gamma, 1 - \gamma]$ then

$$
g(y(1 - a(1 - y))) - g(y) \overset{(32)}{=} \log\left(1 - \frac{a}{1 + ay}\right) \leqslant \log\left(1 - \frac{a}{1 + y}\right) \leqslant \log\left(1 - \frac{a}{2}\right) \leqslant -\frac{a}{2},
$$

and so **B4** holds. If $a \in [0, \frac{1}{2}]$ and $y \in [\gamma, 1 - \gamma]$ then

$$
g(y(1 - a(1 - y))) - g(y) \overset{(32)}{=} \log\left(1 - \frac{a}{1 + ay}\right) \geqslant \log(1 - a) \geqslant -2a,
$$

and so **B5** holds. Finally, if $\gamma \leqslant y \leqslant \frac{1}{2}$ then

$$
-1 - \log y < \log(1/2) - \log y \leqslant \log(1 - y) - \log y \leqslant -\log y.
$$

Observing that $g(1 - y) = \log(1 - y) - \log y$ then confirms **B6**, thus proving the lemma. $\qquad \square$

The following lemma collects some properties of the map $\phi$ that will be useful in our analysis.

**Lemma D.14.** *Suppose that $p \in A_{i-1}$ and $q \in \phi(p)$. Then the following properties hold.*

**C1** *If $p(a_i, c_{i-1}) \geqslant \min\left\{1 - \frac{c}{100}, p(b_i, c_{i-1})\right\}$ and $p(c_i, b_i) \geqslant \frac{1}{8}$, then $q(c_i, b_i) \geqslant \frac{1}{8}$.*

**C2** *If $p(a_i, c_{i-1}) \geqslant 1 - \frac{c}{100}$ and $p(c_i, b_i) \leqslant \frac{1}{8}$, then $g(q(c_i, b_i)) \geqslant g(p(c_i, b_i)) + \frac{c}{32}$.*

**C3** *If $p(c_i, a_i) \geqslant \frac{1}{8}$ and $p(a_i, c_{i-1}) \leqslant 1 - \frac{1}{50n}$, then $g(q(a_i, c_{i-1})) \geqslant g(p(a_i, c_{i-1})) + \frac{1}{64}$.*

**C4** *If $p(c_i, a_i) \geqslant \frac{1}{8}$ then $q(a_i, c_{i-1}) \geqslant p(a_i, c_{i-1})$.*

**C5** *$g(q(a_i, c_{i-1})) \geqslant g(q(a_i, c_{i-1})) - 2c$.*

**C6** *If $p(a_i, c_{i-1}) \leqslant \frac{1}{8}$, then $g(q(a_i, c_{i-1})) \geqslant g(p(a_i, c_{i-1})) + \frac{c}{40}$.*

**C7** *If $p(a_i, c_{i-1}) \geqslant \frac{1}{8}$, then $g(q(a_i, c_{i-1})) \geqslant \frac{1}{8}$.*

**C8** *For some fixed function $h : (0, 1) \to (0, 1)$ that it strictly increasing, $q(c_i, a_i) \geqslant h(p(c_i, a_i))$.*

**C9** *If $p(c_i, b_i) \leqslant \frac{1}{8}$ then $q(c_i, a_i) \geqslant \frac{1}{8}$.*

*Proof.* Let $q' \in \mathcal{Q}_0$ be chosen such that we have

$$
q'(u, v) \geqslant p(u, v) \cdot \left[1 + (\alpha(p; u, v) + c(\tfrac{1}{5} - p(u, v))) \cdot (1 - p(u, v))\right] \quad \text{for every } u, v \in V, \tag{33}
$$

$$
\bar{\pi}_\gamma(q') = q. \tag{34}
$$

**C1:** If $p(a_i, c_{i-1}) \geqslant \min\{1 - \frac{c}{100}, p(b_i, c_{i-1})\}$ then

$$q'(c_i, b_i) \overset{(33)}{\geqslant} p(c_i, b_i) \cdot [1 + (s_{1/2}^+(p(a, c_{i-1}) - p(b_i, c_{i-1})) + c(\tfrac{1}{4} - p(c_i, b_i))) \cdot (1 - p(c_i, b_i))]$$
$$\geqslant p(c_i, b_i) \cdot [1 + (-\tfrac{c}{100} + c(\tfrac{1}{5} - p(c_i, b_i))) \cdot (1 - p(c_i, b_i))]$$
$$\geqslant p(c_i, b_i) \cdot [1 + c(\tfrac{3}{16} - p(c_i, b_i)) \cdot (1 - p(c_i, b_i))]. \tag{35}$$

If additionally $\frac{1}{8} \leqslant p(c_i, b_i) \leqslant \frac{3}{16}$, it then follows that

$$q(c_i, b_i) \overset{(34)}{=} \hat{\pi}_\gamma(q'(c_i, b_i)) \overset{(35)}{\geqslant} \hat{\pi}_\gamma(p(c_i, b_i)) \geqslant \hat{\pi}_\gamma(\tfrac{1}{8}) = \tfrac{1}{8}.$$

On the other hand, if $\frac{3}{16} \leqslant p(c_i, b_i)$ then

$$q(c_i, b_i) \overset{(34)}{=} \hat{\pi}_\gamma(q'(c_i, b_i)) \overset{(35)}{\geqslant} \hat{\pi}_\gamma(p(c_i, b_i) \cdot [1 - c]) \geqslant \hat{\pi}_\gamma(\tfrac{3}{16} \cdot [1 - c]) \geqslant \hat{\pi}_\gamma(\tfrac{1}{8}) = \tfrac{1}{8}.$$

In either case, we find **C1** holds.

**C2:** If $p(a_i, c_{i-1}) \geqslant 1 - \frac{c}{100}$ and $p(c_i, b_i) \leqslant \frac{1}{8}$, then

$$q(c_i, b_i) \overset{(34)}{=} \hat{\pi}_\gamma(q'(c_i, b_i)) \overset{(35)}{\geqslant} \hat{\pi}_\gamma(p(c_i, b_i) \cdot [1 + \tfrac{c}{16}(1 - p(c_i, b_i))])$$
$$\geqslant p(c_i, b_i) \cdot [1 + \tfrac{c}{16}(1 - p(c_i, b_i))].$$

It then follows from **B1** and **B3** that $g(q(i)) \geqslant g(p(i)) + \frac{c}{32}$, and so **C2** holds.

**C3:** Observe that for any $0 \leqslant y \leqslant 1 - \frac{1}{50n}$ and $a \in [0, \frac{1}{2}]$, we have

$$y \cdot [1 + a(1 - y)] \leqslant y \cdot (\tfrac{3}{2} - \tfrac{1}{2}y) \leqslant (1 - \tfrac{1}{50n})(1 + \tfrac{1}{100n}) \leqslant 1 - \tfrac{1}{100n} = 1 - \gamma, \tag{36}$$

and hence

$$g(\hat{\pi}_\gamma(y \cdot [1 + a(1 - y)])) \overset{(36)}{\geqslant} g(y \cdot [1 + a(1 - y)]) \overset{\textbf{B3}}{\geqslant} g(y) + \tfrac{a}{2}. \tag{37}$$

We have

$$q'(a_i, c_{i-1}) \overset{(33)}{\geqslant} p(a_i, c_{i-1}) \cdot [1 + (\tfrac{1}{2}p(c_i, a_i) + c(\tfrac{1}{5} - p(a_i, c_{i-1}))) \cdot (1 - p(a_i, c_{i-1}))]. \tag{38}$$

If $p(c_i, a_i) \geqslant \frac{1}{8}$ and $p(a_i, c_{i-1}) \leqslant 1 - \frac{1}{50n}$, then

$$g(q(a_i, c_{i-1})) \overset{(34)}{=} g(\hat{\pi}_\gamma(q'(a_i, c_{i-1}))) \overset{(38)}{\geqslant} g(\hat{\pi}_\gamma(p(a_i, c_{i-1}) \cdot [1 + \tfrac{1}{32}(1 - p(a_i, c_{i-1}))]))$$
$$\overset{(37)}{\geqslant} g(p(a_i, c_{i-1})) + \tfrac{1}{64}.$$

**C4:** If $p(c_i, a_i) \geqslant \frac{1}{8}$, then

$$q(a_i, c_{i-1}) \overset{(34)}{=} \hat{\pi}_\gamma(q'(a_i, c_{i-1})) \overset{(38)}{\geqslant} \hat{\pi}_\gamma(p(a_i, c_{i-1})) = p(a_i, c_{i-1}).$$

**C5:** We have

$$g(q(a_i, c_{i-1})) \overset{(34)}{=} g(\hat{\pi}_\gamma(q'(a_i, c_{i-1}))) \overset{(38)}{\geqslant} g(\pi_\gamma(p(a_i, c_{i-1}) \cdot [1 - c(1 - p(a_i, c_{i-1}))]))$$
$$\geqslant g(p(a_i, c_{i-1}) \cdot [1 - c(1 - p(a_i, c_{i-1}))]) \overset{\textbf{B5}}{\geqslant} g(p(a_i, c_{i-1})) - 2c.$$

**C6:** If $p(a_i, c_{i-1}) \leqslant \frac{1}{8}$, then

$$g(q(a_i, c_{i-1})) \overset{(34)}{=} g(\hat{\pi}_\gamma(q'(a_i, c_{i-1}))) \overset{(38)}{\geqslant} g(\hat{\pi}_\gamma(p(a_i, c_{i-1}) \cdot [1 + \tfrac{3c}{40}(1 - p(a_i, c_{i-1}))]))$$
$$= g(p(a_i, c_{i-1}) \cdot [1 + \tfrac{3c}{40}(1 - p(a_i, c_{i-1}))]) \overset{\textbf{B3}}{\geqslant} g(p(a_i, c_{i-1})) + \tfrac{c}{40}.$$

**C7:** First, if $\frac{1}{8} \leqslant p(a_i, c_{i-1}) \leqslant \frac{1}{4}$ then

$$q(a_i, c_{i-1}) \overset{(34)}{=} \hat{\pi}_\gamma(q'(a_i, c_{i-1})) \overset{(38)}{\geqslant} \hat{\pi}_\gamma(p(a_i, c_{i-1})) = p(a_i, c_{i-1}) \geqslant \tfrac{1}{8}.$$

On the other hand, if $\frac{1}{4} \leqslant p(a_i, c_{i-1})$ then

$$q(a_i, c_{i-1}) \stackrel{(34)}{=} \pi_\gamma(q'(a_i, c_{i-1})) \stackrel{(38)}{\geqslant} \pi_\gamma(p(a_i, c_{i-1}) \cdot (1-c)) \geqslant \pi_\gamma(\tfrac{1}{8}) = \tfrac{1}{8}.$$

**C8:** We have

$$q(c_i, a_i) \stackrel{(34)}{=} \hat{\pi}_\gamma(q'(c_i, a_i))$$

$$\stackrel{(33)}{\geqslant} \pi_\gamma(p(c_i, a_i) \cdot [1 + (s_{1/2}^+(p(b_i, c_{i-1}) - p(a_i, c_{i-1})) + c(\tfrac{1}{5} - p(c_i, a_i))) \cdot (1 - p(i))])$$

$$\geqslant \hat{\pi}_\gamma(p(c_i, a_i) \cdot [1 + (c(\tfrac{1}{5} - p(c_i, a_i)) - 1) \cdot (1 - p(c_i, a_i))]) \geqslant p(c_i, a_i) \cdot h(p(c_i, a_i)),$$

where $h : (0,1) \to (0,1)$ is the strictly increasing function given by

$$h(x) = x \cdot [1 + (c(\tfrac{1}{5} - x) - 1) \cdot (1 - x)].$$

(To see that $h$ is strictly increasing, note that we may write

$$\frac{d}{dx} h(x) = [1 + (c(\tfrac{1}{5} - x) - 1) \cdot (1 - x)] + x \cdot [(1 - c(\tfrac{1}{5} - x)) - c(1 - x)]$$

$$= 2x + c \cdot (\tfrac{1}{5} - \tfrac{12}{5}x + 3x^2),$$

and so we have $\frac{d}{dx} h(x) > 0$ for $c$ sufficiently small.)

**C9:** If $p(c_i, b_i) \leqslant \frac{1}{8}$, then $p(c_i, a_i) \geqslant \frac{7}{8}$ and hence

$$q(c_i, a_i) \stackrel{\mathbf{C8}}{\geqslant} h(p(c_i, a_i)) \geqslant h(7/8) = \tfrac{7}{8} \cdot [1 + (-\tfrac{5}{8} - c) \cdot \tfrac{1}{8}] \geqslant \left(\tfrac{7}{8}\right)^2 \geqslant \tfrac{1}{8}.$$

$\square$

The following lemma is essentially the dual of Lemma D.14.

**Lemma D.15.** *Suppose that $p \in A_{i-1}$ and $q \in \phi(p)$. Then the following properties hold.*

**D1** *If $p(b_i, c_{i-1}) \geqslant \min\{1 - \frac{c}{100}, p(a_i, c_{i-1})\}$ and $p(c_i, a_i) \geqslant \frac{1}{8}$, then $q(c_i, a_i) \geqslant \frac{1}{8}$.*

**D2** *If $p(b_i, c_{i-1}) \geqslant 1 - \frac{c}{100}$ and $p(c_i, a_i) \leqslant \frac{1}{8}$, then $g(q(c_i, a_i)) \geqslant g(p(c_i, a_i)) + \frac{c}{32}$.*

**D3** *If $p(c_i, b_i) \geqslant \frac{1}{8}$ and $p(b_i, c_{i-1}) \leqslant 1 - \frac{1}{50n}$, then $g(q(b_i, c_{i-1})) \geqslant g(p(b_i, c_{i-1})) + \frac{1}{64}$.*

**D4** *If $p(c_i, b_i) \geqslant \frac{1}{8}$ then $q(b_i, c_{i-1}) \geqslant p(b_i, c_{i-1})$.*

**D5** $g(q(b_i, c_{i-1})) \geqslant g(q(b_i, c_{i-1})) - 2c$.

**D6** *If $p(b_i, c_{i-1}) \leqslant \frac{1}{8}$, then $g(q(b_i, c_{i-1})) \geqslant g(p(b_i, c_{i-1})) + \frac{c}{40}$.*

**D7** *If $p(b_i, c_{i-1}) \geqslant \frac{1}{8}$, then $g(q(b_i, c_{i-1})) \geqslant \frac{1}{8}$.*

**D8** *For some fixed function $h : (0,1) \to (0,1)$ that it strictly increasing, $q(c_i, b_i) \geqslant h(p(c_i, b_i))$.*

**D9** *If $p(c_i, a_i) \leqslant \frac{1}{8}$ then $q(c_i, b_i) \geqslant \frac{1}{8}$.*

*Proof.* Swap the roles of $a_i$ and $b_i$ in the proof of Lemma D.14. $\square$

An observation that will be required in the proof of Lemma D.8 is the following.

**Lemma D.16.** *For any $i \in [n]$, $\Phi(A_i) \subseteq A_i$.*

*Proof.* Suppose that $q \in \Phi(A_i)$. There exists $p \in A_i$ such that $q \in \phi(p)$. For every $j \in [i]$, we can now deduce that

$$q(c_j, a_j) \stackrel{\mathbf{D1}}{\geqslant} \tfrac{1}{8},$$

$$q(c_j, b_j) \stackrel{\mathbf{C1}}{\geqslant} \tfrac{1}{8},$$

$$q(a_j, c_{j-1}) \stackrel{\mathbf{C4}}{\geqslant} p(a_j, c_{j-1}) \geqslant 1 - \tfrac{1}{50n},$$

$$q(b_j, c_{j-1}) \stackrel{\mathbf{D4}}{\geqslant} p(b_j, c_{j-1}) \geqslant 1 - \tfrac{1}{50n}.$$

Therefore, $q \in A_i$, and hence $\Phi(A_i) \subseteq A_i$. $\square$

*Proof of Lemma D.8.* We will say that a sequence $(q_t)_{t=0}^r$ in $\mathcal{Q}$ is a $\phi$-sequence if $q_t \in \phi(q_{t-1})$ holds for every $t \in [r]$. Given subsets $A, B \subseteq \mathcal{Q}$ and $r \in \mathbb{N}$, we say that $A \hookrightarrow_r B$ if for any $\phi$-sequence $(q_t)_{t=0}^r$ with $q_0 \in A$, there exists some $t$ with $q_t \in B$. Note that the following properties of $\hookrightarrow_r$ hold.

**E1** If $A \hookrightarrow_r B \hookrightarrow_s C$, then $A \hookrightarrow_{r+s} C$.

**E2** If $A \hookrightarrow_r B_1 \cup B_2$ and $B_k \hookrightarrow_s C$ for $k \in [2]$, then $A \hookrightarrow_{r+s} C$.

We will establish that $A_{i-1} \hookrightarrow_{8\tau} A_i$ for every $i \in [n]$. To do this, choose a constant $\beta$ such that

**F1** $\beta$ is a sufficiently small constant depending on $c$ for any subsequent relationships between $\beta$ and $c$ to hold, and

**F2** $n \geqslant n_0$, where $n_0$ is a sufficiently large constant depending on $\beta$ (as well as $c$ and $K$) for any subsequent relationships between $n_0$ and $\beta$ to hold.

Note that such a choice of $\beta$ is possible due to **A3**. Let $i \in [n]$ be fixed, and define the following sets (see Figure 5 for an illustrative guide).

$$B = \{p \in A_{i-1}: \ p(c_i, a_i) \geqslant \tfrac{1}{8}, \ p(c_i, b_i) \geqslant \tfrac{1}{8}, \ p(a_i, c_{i-1}) \geqslant 1 - \tfrac{c}{100}, \quad p(b_i, c_{i-1}) \geqslant 1 - \tfrac{c}{100} \quad \}$$
$$C_1 = \{p \in A_{i-1}: \ p(c_i, a_i) \geqslant \beta, \ p(c_i, b_i) \geqslant \tfrac{1}{8}, \ p(a_i, c_{i-1}) \geqslant 1 - n^{-\beta}, \quad p(b_i, c_{i-1}) \geqslant 1 - \tfrac{c}{100} \quad \}$$
$$D_1 = \{p \in A_{i-1}: \ p(c_i, a_i) \geqslant \tfrac{1}{8}, \ p(c_i, b_i) \geqslant \tfrac{1}{8}, \ p(a_i, c_{i-1}) \geqslant 1 - n^{-1/2}, \ p(b_i, c_{i-1}) \geqslant \tfrac{1}{8} \quad\quad \}$$
$$E_1 = \{p \in A_{i-1}: \ p(c_i, a_i) \geqslant \tfrac{1}{8}, \ p(c_i, b_i) \geqslant \gamma, \ p(a_i, c_{i-1}) \geqslant 1 - n^{-1/2}, \ p(b_i, c_{i-1}) \geqslant 1 - n^{-1/2} \ \}$$
$$F_1 = \{p \in A_{i-1}: \ p(c_i, a_i) \geqslant \tfrac{1}{8}, \ p(c_i, b_i) \geqslant \gamma, \ p(a_i, c_{i-1}) \geqslant \tfrac{1}{8}, \quad\quad p(b_i, c_{i-1}) \geqslant 1 - \tfrac{1}{50n} \quad \}$$
$$G_1 = \{p \in A_{i-1}: \ p(c_i, a_i) \geqslant \gamma, \ p(c_i, b_i) \geqslant \gamma, \ p(a_i, c_{i-1}) \geqslant \tfrac{1}{8}, \quad\quad p(b_i, c_{i-1}) \geqslant 1 - \tfrac{1}{50n} \quad \}$$
$$H = \{p \in A_{i-1}: \ p(c_i, a_i) \geqslant \gamma, \ p(c_i, b_i) \geqslant \gamma, \ p(a_i, c_{i-1}) \geqslant \tfrac{1}{8}, \quad\quad p(b_i, c_{i-1}) \geqslant \tfrac{1}{8} \quad\quad \}$$

$$C_2 = \{p \in A_{i-1}: \ p(c_i, a_i) \geqslant \tfrac{1}{8}, \ p(c_i, b_i) \geqslant \beta, \ p(a_i, c_{i-1}) \geqslant 1 - \tfrac{c}{100}, \quad p(b_i, c_{i-1}) \geqslant 1 - n^{-\beta} \quad \}$$
$$D_2 = \{p \in A_{i-1}: \ p(c_i, a_i) \geqslant \tfrac{1}{8}, \ p(c_i, b_i) \geqslant \tfrac{1}{8}, \ p(a_i, c_{i-1}) \geqslant \tfrac{1}{8}, \quad\quad p(b_i, c_{i-1}) \geqslant 1 - n^{-1/2} \ \}$$
$$E_2 = \{p \in A_{i-1}: \ p(c_i, a_i) \geqslant \gamma, \ p(c_i, b_i) \geqslant \tfrac{1}{8}, \ p(a_i, c_{i-1}) \geqslant 1 - n^{-1/2}, \ p(b_i, c_{i-1}) \geqslant 1 - n^{-1/2} \ \}$$
$$F_2 = \{p \in A_{i-1}: \ p(c_i, a_i) \geqslant \gamma, \ p(c_i, b_i) \geqslant \tfrac{1}{8}, \ p(a_i, c_{i-1}) \geqslant 1 - \tfrac{1}{50n}, \quad p(b_i, c_{i-1}) \geqslant \tfrac{1}{8} \quad\quad \}$$
$$G_2 = \{p \in A_{i-1}: \ p(c_i, a_i) \geqslant \gamma, \ p(c_i, b_i) \geqslant \gamma, \ p(a_i, c_{i-1}) \geqslant 1 - \tfrac{1}{50n}, \quad p(b_i, c_{i-1}) \geqslant \tfrac{1}{8} \quad\quad \}$$

Using the properties established in Lemmas D.14 and D.15, we will show that $A_{i-1} \hookrightarrow_\tau H \hookrightarrow_\tau G_1 \cup G_2$, $B \hookrightarrow_\tau A_i$, and $G_k \hookrightarrow_\tau F_k \hookrightarrow_\tau E_k \hookrightarrow_\tau D_k \hookrightarrow_\tau C_k \hookrightarrow_\tau B$ for $k \in [2]$. (For those instances of depending on $k \in [2]$ we will only show the case $k = 1$, however the case $k = 2$ will also hold analagously.)

$B \hookrightarrow_\tau A_i$: Suppose that $(q_t)_{t=0}^\tau$ is a $\phi$-sequence with $q_0 \in B$. By induction on $t$, the following properties hold for every $0 \leqslant t \leqslant \tau$.

$$q_t(c_i, a_i) \geqslant \tfrac{1}{8},$$
$$q_t(c_i, b_i) \geqslant \tfrac{1}{8},$$
$$g(q_t(a_i, c_{i-1})) \geqslant \min\{g(1 - \tfrac{c}{100}) + \tfrac{t}{128}, g(1 - \tfrac{1}{50n})\},$$
$$g(q_t(b_i, c_{i-1})) \geqslant \min\{g(1 - \tfrac{c}{100}) + \tfrac{t}{128}, g(1 - \tfrac{1}{50n})\}.$$

Indeed, the condition on $q_t(c_i, a_i)$ follows from **D1**, the condition on $q_t(c_i, b_i)$ follows from **C1**, the condition on $q_t(a_i, c_{i-1})$ follows from **C3** and **C4**, and the condition on $q_t(b_i, c_{i-1})$ follows from **D3** and **D4**. Because, using **B2**, $g(1 - \tfrac{c}{100}) + \tfrac{\tau}{128} \geqslant \log(1/\gamma) \geqslant g(1 - \tfrac{1}{50n})$, we then have $q_\tau \in A_i$.

$C_1 \hookrightarrow_\tau B$: Set

$$\tau' := \left\lceil \frac{g(\tfrac{1}{8}) - g(\beta)}{c/32} \right\rceil \leqslant \tau.$$

Suppose that $(q_t)_{t=0}^{\tau'}$ is a $\phi$-sequence with $q_0 \in C_1$. By induction on $t$, the following properties hold for every $0 \leqslant t \leqslant \tau'$.

$$q_t(c_i, a_i) \geqslant \min\{g(\beta) + t \cdot \tfrac{c}{32}, g(\tfrac{1}{8})\},$$
$$q_t(c_i, b_i) \geqslant \tfrac{1}{8},$$
$$g(q_t(a_i, c_{i-1})) \geqslant g(1 - n^{-\beta}) - t \cdot 2c \geqslant g(1 - \tfrac{c}{100}),$$
$$q_t(b_i, c_{i-1}) \geqslant 1 - \tfrac{c}{100}.$$

Indeed, the condition on $q_t(c_i, a_i)$ follows from **D1** and **D2**, the condition on $q_t(c_i, b_i)$ follows from **C1**, the condition on $q_t(a_i, c_{i-1})$ follows from **C5** and the fact that

$$g(1 - n^{-\beta}) - \tau' \cdot 2c \overset{\textbf{B6}}{\geqslant} -1 + \beta \log n - \tau' \cdot 2c \overset{\textbf{F2}}{\geqslant} \tfrac{\beta}{2} \log n \overset{\textbf{B6}}{\geqslant} g(1 - n^{-\beta/2}) \geqslant g(1 - \tfrac{c}{100}),$$

and the condition on $q_t(b_i, c_{i-1})$ follows from **D4**. We then have $q_{\tau'} \in B$.

$D_1 \hookrightarrow_\tau C_1$: Set

$$\tau' := \left\lceil \frac{g(1 - \tfrac{c}{100}) - g(\tfrac{1}{8})}{1/64} \right\rceil \leqslant \tau,$$

and let $h : (0, 1) \to (0, 1)$ be a strictly increasing function so that **C8** holds. Suppose that $(q_t)_{t=0}^{\tau'}$ is a $\phi$-sequence with $q_0 \in D_1$. By induction on $t$, the following properties hold for every $0 \leqslant t \leqslant \tau'$.

$$q_t(c_i, a_i) \geqslant h^t(\tfrac{1}{8}),$$
$$q_t(c_i, b_i) \geqslant \tfrac{1}{8},$$
$$g(q_t(a_i, c_{i-1})) \geqslant g(1 - n^{-1/2}) - t \cdot 2c \geqslant g(1 - n^{-\beta}) \geqslant g(1 - \tfrac{c}{100}),$$
$$g(q_t(b_i, c_{i-1})) \geqslant \min\{g(\tfrac{1}{8}) + t \cdot \tfrac{1}{64}, g(1 - \tfrac{1}{50n})\}.$$

Indeed, the condition on $q_t(c_i, a_i)$ follows from **C8**, the condition on $q_t(c_i, b_i)$ follows from **C1**, the condition on $q_t(a_i, c_{i-1})$ follows from **C5** and the fact that

$$g(1 - n^{-1/2}) - \tau' \cdot 2c \overset{\textbf{B6}}{\geqslant} -1 + \tfrac{1}{2} \log n - \tau' \cdot 2c \geqslant \tfrac{1}{4} \log n \overset{\textbf{B6}}{\geqslant} g(1 - n^{-1/4}) \geqslant g(1 - n^{-\beta}),$$

and the condition on $q_t(b_i, c_{i-1})$ follows from **D3** and **D4**. Because $\tau'$ is defined as a function of $c$, **F1** implies that $h^{\tau'}(\tfrac{1}{8}) \geqslant \beta$. Hence $q_{\tau'} \in C_1$.

$E_1 \hookrightarrow_\tau D_1$: Assume for contradiction that there is a $\phi$-sequence $(q_t)_{t=0}^{\tau}$ with $q_0 \in E_1$ and $q_t \notin D_1$ for all $0 \leqslant t \leqslant \tau$. Let $s \geqslant 0$ be the maximal value of $t$ such that the following properties hold.

$$g(q_0(c_i, b_i)) + t \cdot \tfrac{c}{32} \leqslant g(q_t(c_i, b_i)) < g(\tfrac{1}{8}),$$
$$q_t(a_i, c_{i-1}) \geqslant 1 - n^{-1/2},$$
$$q_t(b_i, c_{i-1}) \geqslant \tfrac{1}{8}.$$

Note that $s$ is well defined because the above properties hold for $t = 0$ (as $q_0 \in E_1 \setminus D_1$). Because $g(q_0(c_i, b_i)) + \tau \cdot \tfrac{c}{32} \geqslant g(\tfrac{1}{8})$, it must hold that $s < \tau$. However, we then have

$$g(q_0(c_i, b_i)) + (s+1) \cdot \tfrac{c}{32} \overset{\textbf{C2}}{\leqslant} g(q_{s+1}(c_i, b_i)) \overset{\textbf{C9}}{\leqslant} g(\tfrac{7}{8})$$
$$q_{s+1}(a_i, c_{i-1}) \overset{\textbf{C4}}{\geqslant} 1 - n^{-1/2}$$
$$q_{s+1}(b_i, c_{i-1}) \overset{\textbf{D7}}{\geqslant} \tfrac{1}{8}.$$

If additionally $g(q_{s+1}(c_i, b_i)) < g(\tfrac{1}{8})$ then we have a contradiction to the maximality of $s$, whereas if $g(\tfrac{1}{8}) \leqslant g(q_{s+1}(c_i, b_i))$ then we have $q_{s+1} \in D_1$ (noting in particular that $g(q_{s+1}(c_i, b_i)) \leqslant g(\tfrac{7}{8})$ implies that $q_{s+1}(c_i, a_i) \geqslant \tfrac{1}{8}$). In either case we obtain a contradiction, and so in fact $E_1 \hookrightarrow_\tau D_1$.

$F_1 \hookrightarrow_\tau E_1$: Assume for contradiction that there is a $\phi$-sequence $(q_t)_{t=0}^{\tau}$ with $q_0 \in F_1$ and $q_t \notin E_1$ for all $0 \leqslant t \leqslant \tau$. Let $s \geqslant 0$ be the maximal value of $t$ such that the following properties hold.

$$q_t(c_i, a_i) \geqslant \tfrac{1}{8},$$
$$g(q_0(a_i, c_{i-1})) + t \cdot \tfrac{1}{64} \leqslant g(q_t(a_i, c_{i-1})) \leqslant \min\{g(q_t(b_i, c_{i-1}), g(1 - \tfrac{1}{50n})\},$$
$$g(q_t(b_i, c_{i-1})) \geqslant g(1 - \tfrac{1}{50n}) - t \cdot 2c.$$

Note that $s$ is well defined because the above properties hold for $t = 0$ (as $q_0 \in F_1 \setminus E_1$). Because $g(q_0(a_i, c_{i-1})) + s \cdot \frac{1}{64} \leqslant g(1 - \frac{1}{50n})$, it must hold that

$$s \overset{\mathbf{B2}}{\leqslant} 2 \log{(1/\gamma)}/(1/64) = 128 \log{(100n)} < \tau. \tag{39}$$

However, we then have

$$q_{s+1}(c_i, a_i) \overset{\mathbf{D1}}{\geqslant} \tfrac{1}{8},$$

$$g(q_0(a_i, c_{i-1})) + (s+1) \cdot \tfrac{1}{64} \overset{\mathbf{C3}}{\leqslant} g(q_{s+1}(a_i, c_{i-1})),$$

$$g(q_{s+1}(b_i, c_{i-1})) \overset{\mathbf{D5}}{\geqslant} g(1 - \tfrac{1}{50n}) - (s+1) \cdot 2c. \tag{40}$$

Therefore, by the maximality of $s$,

$$g(q_{s+1}(a_i, c_{i-1})) > \min{\{g(q_{s+1}(b_i, c_{i-1})), g(1 - \tfrac{1}{50n})\}} \overset{(40)}{\geqslant} g(1 - \tfrac{1}{50n}) - (s+1) \cdot 2c$$

$$\overset{\mathbf{B6},(39)}{\geqslant} (\log{(50n)} - 1) - 257c \log{(100n)}$$

$$\overset{\mathbf{A2}}{\geqslant} \log n + \log 50 - 1 - \tfrac{257}{600} \log n - \tfrac{257}{600} \log 100 \overset{\mathbf{F2}}{\geqslant} \log{(\sqrt{n})} \overset{\mathbf{B6}}{\geqslant} g(1 - n^{-1/2}).$$

But then $q_{s+1} \in E_1$, a contradiction. So in fact $F_1 \hookrightarrow_\tau E_1$.

$G_1 \hookrightarrow_\tau F_1$: Assume for contradiction that there is a $\phi$-sequence $(q_t)_{t=0}^\tau$ with $q_0 \in G_1$ and $q_t \notin F_1$ for all $0 \leqslant t \leqslant \tau$. Let $s \geqslant 0$ be the maximal value of $t$ such that the following properties hold.

$$g(q_0(c_i, a_i)) + t \cdot \tfrac{c}{32} \leqslant g(q_t(c_i, a_i)) < g(\tfrac{1}{8}),$$

$$q_t(a_i, c_{i-1}) \geqslant \tfrac{1}{8},$$

$$q_t(b_i, c_{i-1}) \geqslant 1 - \tfrac{1}{50n}.$$

Note that $s$ is well defined because the above properties hold for $t = 0$ (as $q_0 \in G_1 \setminus F_1$). Because $g(q_0(c_i, a_i)) + \tau \cdot \frac{c}{32} \geqslant g(\frac{1}{8})$, it must hold that $s < \tau$. However, we then have

$$g(q_0(c_i, a_i)) + (s+1) \cdot \tfrac{c}{32} \overset{\mathbf{D2}}{\leqslant} g(q_{s+1}(c_i, a_i)),$$

$$q_{s+1}(a_i, c_{i-1}) \overset{\mathbf{C7}}{\geqslant} \tfrac{1}{8},$$

$$q_{s+1}(b_i, c_{i-1}) \overset{\mathbf{D4}}{\geqslant} 1 - \tfrac{1}{50n}.$$

If additionally $g(q_{s+1}(c_i, a_i)) < g(\frac{1}{8})$ then we have a contradiction to the maximality of $s$, whereas if $g(q_{s+1}(c_i, a_i)) \geqslant g(\frac{1}{8})$ then we have $q_{s+1} \in F_1$. In either case we obtain a contradiction, and so in fact $G_1 \hookrightarrow_\tau F_1$.

$H \hookrightarrow_\tau G_1 \cup G_2$: Assume for contradiction that there is a $\phi$-sequence $(q_t)_{t=0}^\tau$ with $q_0 \in H$ and $q_t \notin G_1 \cup G_2$ for all $0 \leqslant t \leqslant \tau$. Using $\mathbf{C7}$ and $\mathbf{D7}$, we have $\phi(p) \subseteq H$ for every $p \in H$, so we can in fact assume that $q_t \in H \setminus (G_1 \cup G_2)$ for every $0 \leqslant t \leqslant \tau$. Let $h : \mathcal{Q} \to \mathbb{R}$ be given by $h(p) = g(p(a_i, c_{i-1})) + g(p(b_i, c_{i-1}))$. If $q_t(c_i, a_i) \geqslant \frac{1}{8}$, then we have

$$h(q_{t+1}) \overset{\mathbf{C3},\mathbf{D5}}{\geqslant} g(q_t(a_i, c_{i-1})) + \tfrac{1}{64} + g(q_t(b_i, c_{i-1})) - 2c \overset{\mathbf{A2}}{\geqslant} h(q_t) + \tfrac{1}{128},$$

whereas if $q_t(c_i, a_i) < \frac{1}{8}$ then $q_t(c_i, b_i) \geqslant \frac{1}{8}$ and hence we have

$$h(q_{t+1}) \overset{\mathbf{D3},\mathbf{C5}}{\geqslant} g(q_t(a_i, c_{i-1})) - 2c + g(q_t(b_i, c_{i-1})) + \tfrac{1}{64} \overset{\mathbf{A2}}{\geqslant} h(q_t) + \tfrac{1}{128}.$$

From this we can deduce that $h(q_\tau) \geqslant h(q_0) + \frac{\tau}{128}$, a contradiction to the range of $h$. Therefore, $H \hookrightarrow_\tau G_1 \cup G_2$.

$A_{i-1} \hookrightarrow_\tau H$: Suppose that $(q_t)_{t=0}^\tau$ is a $\phi$-sequence with $q_0 \in A_{i-1}$. By induction on $t$, the following properties hold for every $0 \leqslant t \leqslant \tau$.

$$g(q_t(a_i, c_{i-1})) \geqslant \min{\{g(\gamma) + t \cdot \tfrac{c}{40}, g(\tfrac{1}{8})\}},$$

$$g(q_t(b_i, c_{i-1})) \geqslant \min{\{g(\gamma) + t \cdot \tfrac{c}{40}, g(\tfrac{1}{8})\}}.$$

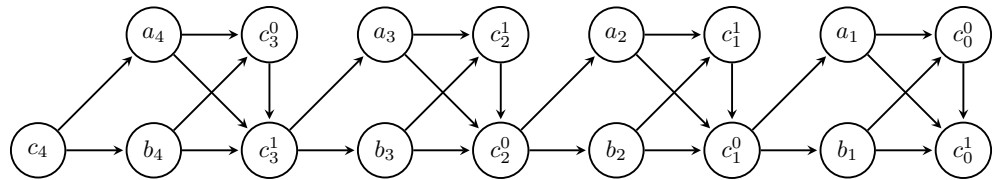

Figure 7: The game RLO(1001).

Indeed, the condition on $q_t(a_i, c_{i-1})$ follows from **C6** and **C7**, and the condition on $q_t(b_i, c_{i-1})$ follows from **D6** and **D7**. Because, using **B2**, $g(\gamma) + \tau \cdot \frac{c}{40} \geqslant \log(1/\gamma) \geqslant g(\frac{1}{8})$, we then have $q_\tau \in H$.

We have now shown that $A_{i-1} \hookrightarrow_\tau H \hookrightarrow_\tau G_1 \cup G_2$, $B \hookrightarrow_\tau A_i$, and $G_k \hookrightarrow_\tau F_k \hookrightarrow_\tau E_k \hookrightarrow_\tau D_k \hookrightarrow_\tau C_k \hookrightarrow_\tau B$ for $k \in [2]$. By applying **E1** and **E2**, we can deduce that $A_{i-1} \hookrightarrow_{8\tau} A_i$ holds for every $i \in [n]$, and hence (by **E1**) that $A_0 \hookrightarrow_{8\tau n} A_n$. Thus, as $A_0 = \mathcal{Q}$, every sequence sequence $(q_t)_{t=0}^{8\tau n}$ in $\mathcal{Q}$ with $q_t \in \phi(q_{t-1})$ for every $t \in [8\tau n]$ must satisfy $q_t \in A_n$ for some $t \in [8\tau n]$. However, using Lemma D.16, this implies that $q_{t+1}, \ldots, q_{8\tau n} \in A_n$ also. Therefore, $\Phi^{8\tau n}(\mathcal{Q}) \subseteq A_n$, as required. $\qquad\square$

# E  Supplementary material to Section 6

## E.1  A generalisation of Reciprocal LeadingOnes

Given a bitstring $z \in \{0,1\}^n$ we define RLO($z$) (Reciprocal LeadingOnes with target $z$) to be the impartial combinatorial game $G = (V, F, v_0)$ with the following definition.

$$V = \{c_n\} \cup \left( \cup_{i \in [n]} \{a_i, b_i, c_{i-1}^0, c_{i-1}^1\} \right)$$

$$v_0 = c_n$$

$$F(v) = \begin{cases} \{a_n, b_n\} & \text{if } v = c_n, \\ \{c_{i-1}^0, c_{i-1}^1\} & \text{if } v = a_i \text{ or } v = b_i, \\ \{c_{i-1}^{z(i)}\} & \text{if } v = c_{i-1}^{1-z(i)}, \\ \{a_i, b_i\} & \text{if } v = c_i^{z(i)} \text{ and } 1 < i \leqslant n, \\ \emptyset & \text{if } v = c_0^{z(i)}. \end{cases}$$

Note that all instances of RLO($z$) are isomorphic to RLO. An example instance is shown in Figure 7.

Given an instance RLO($z$) of Reciprocal LeadingOnes, there is a natural bijection from $\varphi : \{0,1\}^{3n} \to \mathcal{X}_{\text{RLO}(z)}$ given by

$$\varphi(x)(c_i^{z(i+1)}) = \begin{cases} a_i & \text{if } x(i) = 1, \\ b_i & \text{if } x(i) = 0, \end{cases}$$

$$\varphi(x)(a_i) = c_i^{x(n+i)},$$

$$\varphi(x)(b_i) = c_i^{x(2n+i)}.$$

For the sake of a clearer correspondence between our search domain and existing black box models for evolutionary algorithms, and also a neater proof, we will always identify strategies for RLO($z$) using this identification, and hence adopt $\mathcal{X} := \{0,1\}^{3n}$ as the problem's search domain. With this convention, we remark that the discussion of Proposition 3.2 implies that

$$\text{Opt}(\text{RLO}(z)) = \{x \in \mathcal{X} : x(n+i) = x(2n+i) = z(i) \text{ for all } i \in [n]\}.$$

## E.2  Proof of Theorem 6.1

Proving Theorem 6.1 relies on the following preliminaries.

**Lemma E.1.** *Suppose that $z, z' \in \{0,1\}^n$ satisfy $z(i) = z'(i) = 1$ whenever $i \leqslant m$. Let $x \in \mathcal{X}$ and $y \sim \text{Unif}(\mathcal{X})$. Then,*

$$\mathbb{P}(f_{\text{RLO}(z)}(x, y) \neq f_{\text{RLO}(z')}(x, y)) \leqslant 2^{-m}. \tag{41}$$

*Proof.* Observe that for each $w \in \{z, z'\}$ and $j \in \{0, \ldots, m\}$, exactly one player out of $x$ or $y$ takes the turn at position $c_j^1$ in a play of $\mathrm{RLO}(w)$ using $x$ and $y$ (where we here interpret losing due to being unable to move at $c_0^1$ as 'taking a turn'). Let us use $X_j(w)$ to denote the event that this turn is taken by $x$ and $Y_j(w)$ to denote the event that this turn is taken by $y$. Finally, let us define the event

$$E_j = (X_j(z) \wedge Y_j(z')) \vee (Y_j(z) \wedge X_j(z')),$$

and note that $E_0$ occurs if and only if $f_{\mathrm{RLO}(z)}(x, y) \neq f_{\mathrm{RLO}(z')}(x, y)$.

If for some $j \in \{1, \ldots, m\}$ we find that $E_j$ does not occur, then the plays of $\mathrm{RLO}(z)$ and $\mathrm{RLO}(z')$ have coalesced, and so $E_{j-1}$ also does not occur. In particular, we have

$$E_j = \wedge_{i=j}^m E_i \qquad \text{whenever } 0 \leqslant j \leqslant m, \tag{42}$$

and hence,

$$\mathbb{P}(E_0) \overset{(42)}{=} \mathbb{P}(\wedge_{j=0}^m E_j) = \mathbb{P}(E_m) \cdot \prod_{j=1}^m \mathbb{P}(E_{m-j} \mid \wedge_{i=m-j+1}^m E_i)$$

$$\overset{(42)}{=} \mathbb{P}(E_m) \cdot \prod_{j=1}^m \mathbb{P}(E_{m-j} \mid E_{m-j+1}) \leqslant \prod_{j=1}^m \mathbb{P}(E_{m-j} \mid E_{m-j+1}) = \prod_{j=1}^m \mathbb{P}(E_{j-1} \mid E_j).$$

Finally, observe that

$$\mathbb{P}(E_{j-1} \mid E_j) = \mathbb{P}(y(x(c_j^1)) = x(y(c_j^1))) = \sum_{v \in \{a_j, b_j\}} \mathbb{P}(y(x(c_j^1)) = x(v)) \cdot \mathbb{P}(y(c_j^1) = v) = \frac{1}{2}.$$

Thus, we deduce that $\mathbb{P}(E_0) \leqslant 2^{-m}$, as required. $\qquad \square$

We will use a coupling to prove the desired result (for a general background on coupling in the context of evolutionary computing, see [3, 8]). For this, we will first require the notion of total variation and the coupling lemma, as follows. (Note that these notions are usually described in terms of the *distributions* of the variables, rather than the variables themselves as we have done here for conciseness.)

**Definition E.2.** *Let $S$ be a countable set, and let $X_1$ and $X_2$ be $S$-valued random variables. The total variation distance between $X_1$ and $X_2$ is defined as*

$$d_{\mathrm{TV}}(X_1, X_2) = \max_{A \subseteq S} |\mathbb{P}(X_1 \in A) - \mathbb{P}(X_2 \in A)|.$$

The following lemma is Proposition 4.7 of [30].

**Lemma E.3.** *Let $S$ be a countable set, and let $X_1$ and $X_2$ be $S$-valued random variables. Then the following properties hold.*

**G1** $d_{\mathrm{TV}}(X_1, X_2) = \inf \{\mathbb{P}(\hat{X}_1 \neq \hat{X}_2 : (\hat{X}_1, \hat{X}_2)$ *is a coupling of $X_1$ and $X_2$*}.

**G2** *There exists a coupling $(\hat{X}_1, \hat{X}_2)$ of $X_1$ and $X_2$ such that $d_{\mathrm{TV}}(X_1, X_2) = \mathbb{P}(\hat{X}_1 \neq \hat{X}_2)$.*

It is well-known (see Proposition 4.2 of [30]) that the total variation distance satisfies

$$d_{\mathrm{TV}}(X_1, X_2) = \frac{1}{2} \sum_{s \in S} |\mathbb{P}(X_1 = s) - \mathbb{P}(X_2 = s)|. \tag{43}$$

In particular, **G2** implies the existence of a coupling $(\hat{X}_1, \hat{X}_2)$ of $X_1$ and $X_2$ such that

$$\mathbb{P}(\hat{X}_1 = \hat{X}_2) \geqslant 1 - d_{\mathrm{TV}}(X_1, X_2) \overset{(43)}{=} 1 - \frac{1}{2} \sum_{s \in S} |\mathbb{P}(X_1 = s) - \mathbb{P}(X_2 = s)|$$

$$= 1 - \frac{1}{2} \sum_{s \in S} \left( \sup_{i \in [2]} \mathbb{P}(X_i = s) - \inf_{i \in [2]} \mathbb{P}(X_i = s) \right)$$

$$= 1 - \frac{1}{2} \sum_{s \in S} \left( \mathbb{P}(X_1 = s) + \mathbb{P}(X_2 = s) - 2 \inf_{i \in [2]} \mathbb{P}(X_i = s) \right)$$

$$= \sum_{s \in S} \inf_{i \in [2]} \mathbb{P}(X_i = s).$$

We require a generalisation of **G2** that extends this observation to larger families of random variables, as follows (see Theorem 4.2 of [46]).

**Lemma E.4.** *Let $S$ be a countable set. Suppose that $(X_i)_{i \in I}$ is a family of $S$-valued random variables. Then there is a coupling $(\hat{X}_i)_{i \in I}$ of $(X_i)_{i \in I}$ such that $\mathbb{P}(\wedge_{i,j \in I} \hat{X}_i = \hat{X}_j) \geqslant \sum_{s \in S} \inf_{i \in I} \mathbb{P}(X_i = s)$.*

The following key lemma for Theorem 6.1 may apply to other settings, and may also generalise to uncountable $S \subseteq \mathbb{R}$.

**Lemma E.5.** *Suppose that $I$ is a finite indexing set, $S \subseteq \mathbb{R}$ is countable, $(f_i)_{i \in I}$ is a family of stochastic payoff functions $\mathcal{X} \to \mathcal{P}(S)$, and $(B_i)_{i \in I}$ is a family of pairwise disjoint subsets of $\mathcal{X}$. Suppose that $\varepsilon > 0$ satisfies*

$$\sum_{s \in S} \inf_{i \in I} \mathbb{P}(f_i(x) = s) \geqslant 1 - \varepsilon \qquad \text{for every } x \in \mathcal{X}.$$

*Let $\mathcal{A}$ be any instance of Algorithm 2. It then holds for any $\tau \geqslant 0$ that*

$$\frac{1}{|I|} \sum_{i \in I} \mathbb{P}[T_{\mathcal{A}}^{f_i}(B_i) < \tau] \leqslant \tau \cdot \left( \frac{1}{|I|} + \varepsilon \right).$$

*Proof.* For every $x \in \mathcal{X}$, we have $\sum_{s \in S} \inf_{i \in I} \mathbb{P}(f_i(x) = s) \geqslant 1 - \varepsilon$. Thus, using Lemma E.4, for each $t \in \mathbb{N}$ and $x \in \mathcal{X}$, let $(\hat{f}_i(t, x))_{i \in I}$ be a coupling of the family of random variables $(f_i(x))_{i \in I}$ satisfying $\mathbb{P}(\wedge_{i,j \in I} \hat{f}_i(t, x) = \hat{f}_j(t, x)) \geqslant 1 - \varepsilon$.

We will simulate $|I|$-many parallel runs of $\mathcal{A}$, one on each stochastic function in $(f_i)_{i \in I}$, which are carefully coupled together. To do this, we first perform the following steps independently of each other.

- Sample $\hat{x}_0 \sim d_0$.

- For all $(t; i; x) \in \mathbb{N} \times I \times \mathcal{X}$, sample $\hat{a}_t^i(x) \sim \hat{f}_i(t, x)$.

- For all $(t; x_0, \ldots, x_t; a_0, \ldots, a_t) \in \mathbb{N} \times \mathcal{X}^{t+1} \times S^{t+1}$, sample

$$\hat{x}_{t+1}(x_0, \ldots, x_t; a_0, \ldots, a_t) \sim v_t(x_0, \ldots, x_t; a_0, \ldots, a_t).$$

Given $i \in I$, the sequence $(x_t^i)_{t \geqslant 0}$ is then defined recursively via

$$\begin{aligned} x_0^i &= \hat{x}_0, \\ a_t^i &= \hat{a}_t^i(x_t^i), \\ x_{t+1}^i &= \hat{x}_{t+1}(x_0^i, \ldots, x_t^i; a_0^i, \ldots, a_t^i). \end{aligned}$$

The result is that each $(x_t^i)_{t \geqslant 0}$ is identical in distribution to a run of $\mathcal{A}$ on $f_i$.

Given $\tau$, let $E_\tau$ be the event that the sequences $(a_t^i)_{t < \tau}$ are identical for $i \in I$, so that $\mathbb{P}(E_0) = 1$. Note that if $E_\tau$ occurs, then in fact the sequences $(x_t^i)_{t \leqslant \tau}$ are also identical, to $(\overline{x}_t)_{t \leqslant \tau}$ say, for $i \in I$. Thus,

$$\mathbb{P}(E_{\tau+1} \mid E_\tau) \geqslant \mathbb{P}(\wedge_{i,j \in I} \hat{a}_t^i(\overline{x}_\tau) = \hat{a}_t^j(\overline{x}_\tau)) = \mathbb{P}(\wedge_{i,j \in I} \hat{f}_i(\tau, \overline{x}_\tau) = \hat{f}_j(\tau, \overline{x}_\tau)) \geqslant 1 - \varepsilon.$$

Thus, we deduce that $\mathbb{P}(E_\tau) \geqslant (1 - \varepsilon)^\tau \geqslant 1 - \tau\varepsilon$. Because the sets $(B_i)_{i \in I}$ are disjoint, in the event that $E_{\tau-1}$ occurs, at most $\tau$ of the parallel runs can have hit their objective $B_i$. Therefore, if $C_\tau$ is the random variable counting the number of parallel runs that have hit their objective before time $\tau$, we obtain

$$\frac{1}{|I|} \sum_{i \in I} \mathbb{P}[T_{\mathcal{A}}^{f_i}(B_i) < \tau] = \frac{\mathbb{E}[C_\tau]}{|\mathcal{Z}|} = \frac{\mathbb{P}(E_{\tau-1}) \cdot \tau + (1 - \mathbb{P}(E_{\tau-1})) \cdot |I|}{|I|}$$

$$\leqslant \frac{\tau}{|I|} + \tau\varepsilon = \tau \cdot \left( \frac{1}{|I|} + \varepsilon \right),$$

as required. $\qquad\square$

With the tools we have introduced, proving Theorem 6.1 is now easy. We use Lemma E.1 to establish that, with high probability, a run of $\mathcal{A}$ on $\mathrm{RLO}(\mathbf{1}^n)$ will be indistinguishable from a run of $\mathcal{A}$ on $\mathrm{RLO}(z)$ if $z$ and $\mathbf{1}^n$ agree on the final $n/3$ bits. Thus, there is no real reason to find the optimal strategy for $\mathrm{RLO}(\mathbf{1}^n)$ as opposed to the optimal strategy for many other instances of RLO that appear similar against random strategies.

*Proof of Theorem 6.1.* Let

$$\mathcal{Z} = \{z \in \{0,1\}^n : z(i) = 1 \text{ whenever } i \leqslant n/3\},$$

so that $|\mathcal{Z}| = 2^{2n/3}$. For any $x \in \mathcal{X}$ and $z, z' \in \mathcal{Z}$, by noting that sampling $y \sim \mathrm{Unif}(\mathcal{X})$ produces a coupling $(f_{\mathrm{RLO}(z)}(x,y), f_{\mathrm{RLO}(z')}(x,y))$ of $(g_z(x), g_{z'}(x))$, we have

$$d_{\mathrm{TV}}(g_z(x), g_{z'}(x)) \overset{\mathbf{G1}}{\leqslant} \mathbb{P}(f_{\mathrm{RLO}(z)}(x,y) \neq f_{\mathrm{RLO}(z')}(x,y)) \overset{(41)}{\leqslant} 2^{-n/3}. \qquad (44)$$

Given $k \in \{-1, 1\}$ and $x \in \mathcal{X}$, let $z(k,x) \in \mathcal{Z}$ satisfy $\mathbb{P}(f_{z(k,x)}(x) = k) = \inf_{z \in \mathcal{Z}} \mathbb{P}(f_z(x) = k)$. We now have for every $x \in \mathcal{X}$ that

$$\sum_{k \in \{-1,1\}} \inf_{z \in \mathcal{Z}} \mathbb{P}(g_z(x) = k) = \mathbb{P}(g_{z(x,1)}(x) = 1) + \mathbb{P}(g_{z(x,-1)}(x) = -1)$$

$$= 1 + \mathbb{P}(g_{z(x,1)}(x) = 1) - \mathbb{P}(g_{z(x,-1)}(x) = 1)$$

$$\geqslant 1 - d_{\mathrm{TV}}(g_{z(x,1)}(x), g_{z(x,-1)}(x)) \overset{(44)}{\geqslant} 1 - 2^{-n/3}.$$

Therefore, we can apply Lemma E.5 with $\varepsilon = 2^{-n/3}$ to obtain.

$$\frac{1}{|\mathcal{Z}|} \sum_{z \in \mathcal{Z}} \mathbb{P}[T_{\mathcal{A}}^{g_z}(\mathrm{Opt}(g_z)) < 2^{n/8}] \leqslant 2^{n/8} \cdot (2^{-2n/3} + 2^{-n/3}) \leqslant 2^{-n/8}.$$

In particular, there is some $z \in \{0,1\}^n$ such that $\mathbb{P}[T_{\mathcal{A}}^{g_z}(\mathrm{Opt}(g_z)) < 2^{n/8}] \leqslant 2^{-n/8}$, as required. $\qquad \square$

## E.3  Proof of Corollary 6.2

We quickly outline some further notation for this proof. $\mathrm{Sym}([3n])$ will be used to denote the set of bijections $\sigma : [3n] \to [3n]$. We recall that $\mathcal{X} := \{0,1\}^{3n}$ (as per Section 6), and additionally use $\oplus$ to denote the exclusive or-operation on bitstrings, and $\sigma_b : \mathcal{X} \to \mathcal{X}$ to denote the permutation over bitstrings associated to $\sigma$, defined via $\sigma_b(x)(i) = x(\sigma(i))$. Finally, we say that real-valued random variables $A$ and $B$ are *equal in distribution*, written $A \overset{d}{=} B$ if $\mathbb{P}(A \leqslant x) = \mathbb{P}(B \leqslant x)$ holds for every $x \in \mathbb{R}$.

We adopt the following definition of unbiased variation [28].

**Definition E.6.** *A variation operator $v : \mathcal{X}^k \times \mathbb{R}^k \to \mathcal{P}(\mathcal{X})$ is* unbiased *if the following conditions are satisfied for all $x_1, \ldots, x_k, y \in \mathcal{X}$ and $a_1, \ldots, a_k \in \mathbb{R}$.*

**H1** *For all $z \in \mathcal{X}$,*

$$v(x_1, \ldots, x_k; a_1, \ldots, a_k)(y) = v(x_1 \oplus z, \ldots, x_k \oplus z; a_1, \ldots, a_k)(y \oplus z).$$

**H2** *For all $\sigma \in \mathrm{Sym}([3n])$,*

$$v(x_1, \ldots, x_k; a_1, \ldots, a_k)(y) = v(\sigma_b(x_1), \ldots, \sigma_b(x_k); a_1, \ldots, a_k)(\sigma_b(y)).$$

We are now ready to prove Corollary 6.2.

*Proof of Corollary 6.2.* Given a bitstring $w \in \mathcal{X}$ and a set $B \subseteq \mathcal{X}$, let us define $B \oplus w \subseteq \mathcal{X}$ via

$$B \oplus w = \{x \oplus w : x \in B\}.$$

Furthermore, Given a bitstring $w \in \mathcal{X}$ and a function $g : \mathcal{X} \to S$, let us define $g \oplus w : \mathcal{X} \to \mathbb{R}$ via

$$(g \oplus w)(x) = g(x \oplus w).$$

Note that for any set $B \subseteq \mathcal{X}$, function $g : \mathcal{X} \to \mathbb{R}$, and bitstring $w \in \mathcal{X}$ we have

$$T_{\mathcal{A}}^g(B) \overset{d}{=} T_{\mathcal{A}}^{g \oplus w}(B \oplus w). \tag{45}$$

By Theorem 6.1 there exists $\overline{z} \in \{0,1\}^n$ such that $\mathbb{P}[T_{\mathcal{A}}^{g_{\overline{z}}}(\mathrm{Opt}(\mathrm{RLO}(\overline{z}))) < 2^{n/8}] \leqslant 2^{-n/8}$. For any $z \in \{0,1\}^n$, let $w_z \in \mathcal{X}$ be defined by

$$w_z(kn + i) = \begin{cases} 0 & \text{if } k = 0, \\ 0 & \text{if } k \in \{1,2\} \text{ and } z(i) = \overline{z}(i), \\ 1 & \text{if } k \in \{1,2\} \text{ and } z(i) \neq \overline{z}(i). \end{cases}$$

Observe that $g_z = g_{\overline{z}} \oplus w_z$ and $\mathrm{Opt}(\mathrm{RLO}(z)) = \mathrm{Opt}(\mathrm{RLO}(\overline{z})) \oplus w_z$. Hence,

$$\mathbb{P}[T_{\mathcal{A}}^{g_z}(\mathrm{Opt}(\mathrm{RLO}(z))) < 2^{n/8}] = \mathbb{P}[T_{\mathcal{A}}^{g_{\overline{z}} \oplus w_z}(\mathrm{Opt}(\mathrm{RLO}(z)) \oplus w_z) < 2^{n/8}]$$

$$\overset{(45)}{=} \mathbb{P}[T_{\mathcal{A}}^{g_{\overline{z}}}(\mathrm{Opt}(\mathrm{RLO}(\overline{z}))) < 2^{n/8}] \leqslant 2^{-n/8},$$

as required. $\qquad\square$

