# OpenReview forum: "Why Playing Against Diverse and Challenging Opponents Speeds Up Coevolution: A Theoretical Analysis on Combinatorial Games"
_NeurIPS.cc/2025/Conference — NeurIPS 2025 poster_

### Official Review · Reviewer_7KRp · 2025-06-18

**Clarity:** 3
**Significance:** 3
**Originality:** 3
**Rating:** 5
**Confidence:** 3

**Summary:**

The main contribution of this paper is to advance the theoretical understanding of coevolution by undertaking a more detailed runtime analysis of CoEAs on combinatorial games, which are a challenging domain for CoEAs. Through this detailed analysis, it proves an improved upper bound, as well as a new lower bound that shows the result is tight up to a polylogarithmic factor. It also shows that an EA cannot efficiently optimise Reciprocal LeadingOnes if opponents are sampled purely uniformly at random.

**Questions:**

To what extent can the findings (other than Theorem 1.2/6.1, which applies generally) be generalized or extended to games beyond RLO? Would doing so require substantially different proof techniques?

Do you currently have any theoretical insight on the c parameter of the Mutate operator (lines 202-212), or only empirical results?

How well do your results generalize to the non-binary case (lines 119-121)?

**Ethical Concerns:**

["NO or VERY MINOR ethics concerns only"]

**Final Justification:**

The authors have answered my questions to my satisfaction. Although it would be nice to go beyond RLO, that setting is significant enough on its own, as the authors explained in their response to Reviewer 97ie. I encourage the authors to include this explanation in the paper for motivation. I lean toward acceptance.

**Limitations:**

yes

**Quality:**

3

**Strengths And Weaknesses:**

The idea presented in the paper seems sound. The exposition is mostly clear. Its findings are relevant to the field of CoEA and appear to be novel.

Page-by-page comments and suggestions:

Page 1:

4: associated to -> associated with

5: provide -> provide a

28-32: Split into separate sentences

Page 2:

43: and so we consider this as a -> so we consider this a

52: Add comma after "even in such cases"

82-83: Put AlphaStar comment in new sentence

Page 3:

87: Add comma after "In this paper"

88-89: Replace semicolon with period (new sentence)

89: "a method of ranking diversity" Briefly elaborate.

119: Make "however..." a new sentence

120: "we only use that when |S_i| = 2 the constraint..." grammatically awkward, reword

122: difference from -> difference to

134: Put quotes around 'balanced' and 'stable', since these are technical terms that are being introduced

Page 4:

135: Suggest clarifying explicitly what the expectation is with respect to

138: Suggest adding a brief informal description of LeadingOnes here

148: state -> describe

153: empty the -> empty, the

Page 5:

Move Figure 1 to this page, if possible

Page 6:

Figure 3: Consider adding error bars (e.g. standard error of the mean)

190: Period before "however" (new sentence)

194-195: Broken citations

208: What is remarkable is that -> It is remarkable that,

Page 8:

Figure 5: Font size for the axes labels is too small

252: who avoids -> avoids

264-265: What precisely do you mean by "is not guaranteed to exist in general"?

265: Add comma before "there is"

Page 9:

294: Add comma after "In addition"

300: which has here been generalised -> which, here, has been generalised

307: to to -> to

---

> ### Author Rebuttal · Authors · 2025-07-31
>
> Thank you very much for your detailed review and your many helpful suggestions to improve clarity, readability, and grammar.
>
> **To what extent can Theorem 1.2/5.2 be generalised or extended?** This is an important question. The overall proof technique is certainly not exclusive to the specific structure of RLO, and there is no innate reason it could not be applied to other combinatorial games. However, one way RLO simplifies our analysis is due to the fact it consists of a sequence or repeating “gadgets” (see Figure 1). Essentially, the vast majority of the proof analyses the dynamics on a single such gadget. If the structure of the game were more complex or varied then the same argument could be recreated, however this may be impractical given how involved the proof process is even for RLO. (We would also like to emphasise that there are valuable insights to be derived from analysing RLO itself, as outlined in the reply to Reviewer 97ie).
>
> **Theoretical insights into the parameter c.** Having c>0 plays an important and enabling role in our proof. The benefit given to us is explained at a very high level in lines 274-282 in the context of the overall proof idea, and its impact is described mathematically in Lemma E.3 (on which many critical steps in the proof ultimately rely). Our main result indicates that near best-possible performance is attained provided c is a sufficiently small constant, however we were not able to quantify theoretically exactly how small and whether there is an optimal theoretical parameter setting.
>
> **Generalisation to non-binary case.** This is another great question! We believe the proof should generalise fairly straightforwardly to the non-binary case, but with some additional details. Where the present proof focuses on $p(u,c_{i-1})$ (where $u\in\{a_i,b_i\}$), one would instead need to focus on $\sum_{v\in W(u)}p(u,v)$, where $W(u)$ is the set of winning moves that can be made from $u$. As this is just a linear combination of the $p(u,v)$, many of the required bounds and equations follow just the same as with the existing analysis. One added complication is that in the present proof we need to show that (with the stabilising operator) players become on average “indifferent” between 2 losing moves; extending this part of the argument to more moves would require some careful analysis of both the constraint function and Mutate procedure. Nonetheless, ultimately the reason we chose to restrict attention to a binary case was to simplify notation and make the proof clearer.

---

> ### Comment · Reviewer_7KRp · 2025-08-04
>
> Thank you for answering my questions. I encourage you to incorporate the explanation you gave to Reviewer 97ie about the significance of RLO in the paper, for motivation. It would be especially helpful to explicitly explain/discuss, at least briefly, how your results could be extended beyond RLO, and what obstructions or theoretical difficulties there might be to doing so. Could you do that?

---

> > ### Author Response · Authors · 2025-08-07
> >
> > Yes, we would be glad to incorporate the discussions here about the significance of RLO and the scope for extending results beyond RLO into the main content of the paper. Thank you for the guidance and feedback.

---

### Official Review · Reviewer_ZkQQ · 2025-06-30

**Clarity:** 3
**Significance:** 3
**Originality:** 4
**Rating:** 5
**Confidence:** 3

**Summary:**

The paper continues the line of work that analyse self-playing Univariate Marginal Distribution Algorithm in combinatorial games. A game, whose optimal strategies are characterized by a sequence of independent "correct" moves, is introduced as a testbed to analyse the dynamic of coevolution. The analysis provides insight into the utility of maintaining diverse opponents in finding marginal move distributions aligned with optimal strategies. This insight is demonstrated empirically.

**Questions:**

Theorem 1.1 summarizes the upper bound as $O(n^2\log^3n)$. This means that $C_{pop}$ in Theorem 5.2 is $O(1)$. It is not clear how the assumption of Theorem 5.2 implies this. Is it not possible that $C_{pop}$ is $\Theta(n)$ or $\Theta(\log n)$?

**Ethical Concerns:**

["NO or VERY MINOR ethics concerns only"]

**Final Justification:**

The authors addressed the concern that the implication of the main theorem can be misinterpreted as the novel relation is not sufficiently explained in the main text, while simultaneously offers a very loose coupling between the operands. While the latter strengthens the claim itself, it can obfuscate the insight on parameter selection. Assuming that this clarity issue will be addressed in a subsequent revision, I will keep my score.

**Limitations:**

yes

**Quality:**

3

**Strengths And Weaknesses:**

The studied problem is significantly more challenging than optimizing LeadingOnes since there is no measure of "progress" towards optimal play. Rather, the algorithm is responsible for maintaining this "measure" via self-play, in the form of adversarial strategies. The simplicity of the combinatorial game, thus, helps expose the effect of opponent distributions on the discovery of best strategies. As such, the study gives meaningful insight into the self-play dynamics, which is reminiscent of the bias-variance trade-off phenomenon in reinforcement learning.

The theoretical findings are summarized in two theorems. The second one is particularly useful since it applies to general black-box searchers. On the other hand, the first one is conditioned on rather stringent parameter settings and the use of the novel mutation operator. A more fine-grained analysis of the effect of the mutation's parameter $c$ on the discovery time distribution would be appreciated.

The proof details are very involved, and not described in a self-contained manner. Following the proofs is challenging as the result.

Some minor things:

One of the two experiments in the paper is not presented with enough information. Specifically the one on the RLO: the number of runs is not given and the variances are not shown. Adding log-scaled variance whiskers to the points in Figure 3 should suffice.

Figure 4: two more trajectories (one for each subfigure) should probably be added whose starting point matches the initial distribution $p_0$ in Algorithm 1.

Figure 6: perhaps the overall runtimes can be added to the second set of images to contextualize the horizontal scaling.

Rasterized images are used in figures. It is advisable to use vectorized images instead.

---

> ### Author Rebuttal · Authors · 2025-07-31
>
> Thank you very much for your helpful and detailed review. We are very pleased you recognise the challenge inherent in the studied problem and the usefulness of the result for black-box searchers.
>
> To answer your question about $C_{\text{pop}}$: Indeed, $C_{\text{pop}}$ is $O(1)$ in Theorem 1.1. It is likely that the lack of clarity here has come about from the use of hierarchy notation $\ll$ which, while convenient for concisely stating relations between constants, is not always standardised across all disciplines and sub-disciplines. In the sense we are using it (which is stated formally in Appendix A), the assumption $1/n_0 \ll 1/C_{\text{pop}} \ll c \ll 1/K$ is essentially the same as “for fixed $n_0$, $C_\text{pop}$, $c$, $K$, the following holds provided provided $c$ is sufficiently small (how small depends on $K$), $C$ is sufficiently large (how large depends on $c$), $n_0$ is sufficiently large (how large depends on $n_0$)”. Because the subsequent statement takes $n \geqslant n_0$, all four of these numbers are $O(1)$ as $n\to\infty$. Therefore we certainly cannot allow $C_\text{pop}$ to behave as a function of $n$, so it is not possible for it to be $\Theta(n)$ or $\Theta(\log{n})$. We appreciate this is not obvious at a glance in the current notation, and so we can rewrite the theorem to make $C_\text{pop}=O(1)$ clearer.
>
> Other reviewers have indicated the need for confidence intervals or variance whiskers in Figure 3, and we will gladly add these in any subsequent revisions. As with the other experiments in Section 4, each plotted runtime was calculated using 100 runs (using the population size that yielded best performance), and we will seek to clarify this in the main text. Thank you also for your suggested improvements to the other figures in the paper.

---

> > ### Comment · Reviewer_ZkQQ · 2025-08-05
> >
> > Thanks for the answer. Alas, I'm not entirely convinced that "$C_{pop}$ is not a function of $n$". I'd like a thorough inspection of the assumption of Theorem 5.2.
> >
> > The assumption contains two relations that determine the relation between $C_{pop}$ and $n$:
> > 1) $1/n_0\ll1/C_{pop}$
> > 2) $n\geq n_0$
> >
> > According to the definition of $\ll$, (1) says:
> >
> > "There is a function $f:(0,1]\to(0,1]$ where $0<1/n_0\le f(1/C_{pop})$"
> >
> > If $f$ is identity, then combining (1) and (2) gives $n\ge C_{pop}$, which does not imply that $C_{pop}$ is $O(1)$.
> >
> > It's possible that I missed a part in the definition of $\ll$ that limits the types of function $f$ can be. As far as I can see, it allows $f$ such that the assumption does not eliminate the possibility of $C_{pop}=O(n^p)$ for any $p>0$.
> >
> > I notice that $\ll$ is only defined (in the paper) within an if-else predicate, and never in an isolated statement. However, I don't see how that would affect the above line of reasoning.

---

> > > ### Author Response · Authors · 2025-08-05
> > >
> > > Thank you for your comment containing further clarification. We believe we understand your concern better now.
> > >
> > > You are correct that if $f$ is the function $f(x)=x$, then we could take $C_\text{pop}=n$ if we so wished. However the conclusion of the theorem is then strictly worse than if we had just taken $C_\text{pop}$ to be a constant instead (the constant would need to be sufficiently small depending on $c$ and $K$, however the conclusion is strongest when these are constants too). This is the reason Theorem 5.2's upper bound may be summarised as $O(n^2\log^3{n})$ -- because we are always able to take $K$, $c$, and $C_\text{pop}$ to be constants in Theorem~5.2 to obtain $O(n^2\log^3{n})$ as a runtime with high probability.
> > >
> > > To help clear any confusion our rebuttal may have caused: when we stated in the rebuttal that "we cannot allow $C_\text{pop}$ to behave as a function of $n$", what was meant was "we cannot allow $C_\text{pop}$ to behave as a function of $n$ *without knowing the implicit function $f:(0,1]\to(0,1]$ realising $1/n_0\ll1/C_\text{pop}$*". If it did turn out that one could take the implicit $f$ to be identity, then the theorem would indeed hold even with $n=C_\text{pop}$; but if the implicit $f$ were $f(x)=x^2$, then we would need to have $n\geqslant C_\text{pop}^2$ for the theorem's conclusion hold. The critical point here is that no matter what the unspecified function $f$ is, then the conclusion always holds if $C_\text{pop}$ is a constant and $n\geqslant n_0$, where $n_0$ is a sufficiently large constant depending on $C_\text{pop}$. This is why we can always take $C_\text{pop}=O(1)$ in Theorem 5.2 (hence giving the strongest possible conclusion) without needing to specify $f$ explicitly.
> > >
> > > It appears there is a need to rewrite the conditions of the theorem to improve clarity. One option would be to write the something along the following lines.
> > >
> > > **Theorem 5.2.** Given any constant $K>0$, there exist constants $c\in(0,1]$ and $C_\text{pop},n_0>0$ such that the following holds. Let $\gamma=1/(100n)$, let $\mathcal{A}$ be described by Algorithm~1, and let $G$ be the game graph for RLO. Then, provided $n\geqslant n_0$ and $\mu\geqslant C_{\text{pop}}\cdot n\log{n}$, [concluding equation].
> > >
> > > Hopefully a reader would find it easier to see from this version why we obtain $O(n^2\log^3{n})$ with appropriate parameter settings. The reason we did not adopt this statement in the first place is because it is technically slightly weaker than the version in the paper currently. Specifically, the form written above suggests there is a fixed (unspecified) value of the parameter $c$ needed in Algorithm 1 for the result to hold. In reality, the result holds for all $c$ smaller than this quantity, provided there is an appropriate scaling of the population size $\mu$ and of $n_0$. We believe this is an important detail, especially for applications. A possible middle ground that still captures this detail would be to use something like the following.
> > >
> > > **Theorem 5.2.** There are functions $\overline{c}:(0,\infty)\to(0,1]$, $C_\text{pop}:(0,1]\to(0,\infty)$, and $n_0:(0,1]\to(0,\infty)$ such that the following holds for any constants $K,c>0$ satisfying $c\leqslant\overline{c}(K)$. Let $\gamma=1/(100n)$, let $\mathcal{A}$ be described by Algorithm 1, and let $G$ be the game graph for RLO. Then, provided $n\geqslant n_0(c)$ and $\mu\geqslant C_{\text{pop}}(c)\cdot n\log{n}$, [concluding equation]
> > >
> > > (Proposition D.1 can easily have a similar update. Appendix E can also be revised to minimise, or possibly eliminate altogether, use of $\ll$ notation in favour of explicitly declaring $c$, $C_\text{pop}$, $n_0$ to be constants in this way.)

---

> > > > ### Comment · Reviewer_ZkQQ · 2025-08-05
> > > >
> > > > I appreciate the clarification.
> > > >
> > > > To be clear, I don't have an issue with Theorem 5.2, so I don't think it needs to be changed. The only concern I have is that the possibility of choosing $C_{pop}$ to be $O(1)$ could be obscured by the use of a novel notation ($\ll$), which is only fully explained in the Appendix. I think adding a sentence in the main body (preferably close to the theorem itself) claiming that the assumption is weak enough to allow such a selection should suffice.
> > > >
> > > > Alternatively, we can reduce the usage of $\ll$ in the claim itself, while keeping their appearances in the proof. For example, we might consider removing $c\ll1/K$ and replacing $K$ in the bound with some asymptotic function of $c$. We can also replace $1/n_0\ll1/C_{pop}$ with a relation between $C_{pop}$ and $n$, since $n_0$ does not appear in the bound. I think having fewer variables would better expose the relationship between algorithmic parameters like $\mu$ and $c$.

---

> > > > > ### Author Response · Authors · 2025-08-05
> > > > >
> > > > > Sounds like stating the result with fewer variables and without usage of $\ll$ (but keeping in proofs) as you describe is be the best way to increase clarity here. Thank you for your guidance and feedback with this.

---

### Official Review · Reviewer_97ie · 2025-07-06

**Clarity:** 3
**Significance:** 3
**Originality:** 3
**Rating:** 5
**Confidence:** 3

**Summary:**

The manuscript applies the Estimation-of-Distribution Algorithm Tournament UMDA to a newly introduced combinatorial game, Reciprocal LeadingOnes (RLO). It derives a better upper bound on the expected runtime for solving RLO and reports empirical results on several small-scale games that suggest a performance advantage over baseline methods.

**Questions:**

- Please address the three points raised in the “Weaknesses” section above.

**Ethical Concerns:**

["NO or VERY MINOR ethics concerns only"]

**Final Justification:**

My main initial concern was about the novelty of this work in comparison to reference [3]. The authors' rebuttal has clarified this point to my satisfaction, so I have revised my rating.

**Limitations:**

Yes

**Paper Formatting Concerns:**

There is no problem.

**Quality:**

3

**Strengths And Weaknesses:**

Strengths.
- **Improved theoretical bound** The analysis yields a better upper bound on runtime than earlier results, marking clear analytical progress.

- **Potential broader impact** Although the experiments target relatively simple games, the approach could generalise to more complex domains, mirroring the way EDA-inspired ideas have influenced large-scale systems such as AlphaStar.

Weaknesses.
- **Novelty of the algorithm** Because Tournament UMDA itself was introduced in reference [3], the contribution here appears to be its application and analysis. Please clarify which aspects-parameter settings, proof techniques, or new insights-differ from the original proposal.

- **Status of RLO** The paper seems to be the first to study Reciprocal LeadingOnes. If RLO is indeed novel, please state this explicitly and outline why analysing this game is interesting or representative of a broader problem class.

- **Significance of the bound** The prior bound you improve upon was obtained with the same algorithm in a very recent study [3]. Please elaborate on why this incremental improvement matters-for example, does it reveal new theoretical behaviour or enable practical speed-ups on larger instances?

Minor Comments
- **Comprehensive appendix** The detailed proof in the appendix is valuable. Including a concise, reader-friendly proof sketch in the main text (e.g., an outline of key lemmas and ideas) would make the paper easier to follow without forcing readers to consult supplementary material.

- **Figure 2 readability** The legend font is too small; please enlarge it for clarity.

---

> ### Author Rebuttal · Authors · 2025-07-31
>
> Thank you for your helpful review. We are pleased that you recognise the progress made by our analysis and potential broader impact on large-scale systems. In reply to your request to address the weaknesses set out in your review:
>
> **Novel operator.** The most novel aspect of our approach is the introduction of the operator Mutate (Definition 2.2), which is applied after each selection step in Algorithm 1. This is a novel feature which is not present in the original proposal [3], and a major contribution of this paper is theoretical and empirical insights into the stabilising effect it has on the algorithm.
> Figure 3 and the subsequent discussion highlights the empirical effects on runtime compared to the original proposal (c=0 corresponds to the original proposal, c=0.05 corresponds to use of the novel operator).
> The stabilising effect of the operator has an important and enabling role in the theoretical analysis. This is discussed at a high level in lines 274-282, and has a recurring effect on the proof contained in Appendix E.
> Further discussion and visualisation of the impact of this novel feature which was too detailed for inclusion in the main text is provided in Appendix D.2.
> Our paper also presents novel proof techniques, detailed in our comments about the significance of the bound.
>
> **Novel proof technique.** Our proof requires careful analysis of a non-linear multidimensional stochastic process. As discussed in lines 262-273, existing tools to handle such cases are very limited. We develop a novel proof technique that sequentially restricts the multidimensional space (cf. Figure 5 and the surrounding discussion) that could be applied in other such cases.
>
> **Significance of bound.** The prior bound that we improve upon is a very general bound that applies to arbitrary combinatorial games. However due to this generality of the bound, it is often far weaker than observed performance. Indeed, it was stated in the conclusion of [3] that the proof of the original bound did not account for the accelerated learning due to players capitalising on the mistakes of their opponents, and that the general bound “could be greatly improved through closer analysis of coevolutionary dynamics, especially for specific games”. Our analysis is significant because it demonstrates that by accounting for these interactions one can prove a bound that essentially matches best possible. We also point out that our result includes a lower bound on runtime, whereas [3] only proves an upper bound.
>
> **RLO is archetypical of games that require the retention of learned knowledge.** Whether it’s Chess or Nim, in order to guarantee optimal play in a combinatorial game it is not enough to simply put yourself in a winning position – you must then also maintain the upper hand by responding correctly to any difficult position your opponent could put you in later. This makes learning optimal play difficult, as each play of a game only bears witness to a fraction of the possible positions. It is in this way that learning is improved by training against opponents that are both challenging (so that difficult positions occur) and diverse (so that a *range* of difficult positions occur). RLO captures this notion perfectly – in each play of the game only one of each $a_i$ or $b_i$ can be encountered, yet optimal play necessitates perfect play at both. Like established games it is difficult for coevolutionary algorithms to optimise (see Figure 2), yet the regular structure keeps it amenable to theoretical analysis.

---

> > ### Comment · Reviewer_97ie · 2025-08-08
> >
> > Thank you for your response, which addresses my concerns.
> >
> > - *Novel operator / Novel proof technique:*
> > Thank you for explaining the differences from the original paper [3]. I believe I now understand this point.
> >
> > - *Significance of the bound:* Thank you for the clarification. I no longer have any issues with this point.
> >
> > - *Importance of RLO:* I now also understand the significance of RLO in this context.
> >
> > Based on these clarifications, I will adjust my score accordingly. Thank you again for your thorough explanation.

---

### Official Review · Reviewer_c2zm · 2025-07-19

**Clarity:** 4
**Significance:** 3
**Originality:** 4
**Rating:** 5
**Confidence:** 2

**Summary:**

The authors conduct runtime analysis of an existing competitive co-evolutionary algorithm (CoEA), the univariate marginal distribution algorithm (UMDA), on a specific combinatorial game (Reciprocal LeadingOne, RLO). By introducing a novel mutation operator, for suitably small games the authors show that UMDA's runtime on RLO has an upper bound $\mathcal{O}(n\log^2n)$, improving on the previously known tightest upper bound. The authors also show that their variant of UMDA on RLO has a runtime lower bound of $\mathcal{O}(n^2\log n)$, which is the first lower bound established in this setting. The authors also present a corollary showing that not just population diversity, but also coevolutionary feedback between player and opponent is fundamental to learning.

**Questions:**

* RLO is arguably a very artificial game - to what extent does your approach transfer to games of more practical interest?
* What would be the significance of finding analogous results for EDAs?
* Can you explain - in a practical context - what "small" means wrt game (as you are saying that $\mathcal{O}(\sqrt{n})$ games may have better asymptotic bounds)?

**Ethical Concerns:**

["NO or VERY MINOR ethics concerns only"]

**Final Justification:**

I maintain my high accepting score after carefully engaging with the authors' rebuttal.

**Limitations:**

yes

**Paper Formatting Concerns:**

* tiny labels/captions in Fig 2
* line 23: "interections"
* line 53: "higher that implied"
* line 105: "over the entirely search space"
* line 120: "we only use that when [...] the constraint function"   --> sth about the sentence doesn't seem to work grammar-wise
* line 134: "concepts of [balance] and [stability]"
* line 164: "[the] responder"
* line 307: "analysis to to algorithms"
* line 327: "several of cases"
* D2/line 526: "walk for while"
* D2/line 537: "This is in start contrast'
* E1/line 548: "we will always always assume"

**Quality:**

4

**Strengths And Weaknesses:**

# Strengths

* The paper is of utmost clarity and very well-written.
* The theoretical results, as far as I have been able to verify them in detail, represent significant progress in the chosen setting.
* The empirical results do illustrate practical utility in the authors' chosen mutation operator across a variety of games (Table 1), although it remains unclear whether the results are statistically significant (also see Figure 2).

# Weaknesses

* While I appreciate the novelty of the theoretical results, I am not convinced about the wider theoretical extendability of the authors' findings outside of the chosen UMDA/RLO setting.
* Likewise, I am not entirely convinced whether the authors' presented empirical results (Figure 2) indeed add evidence for the practical utility of their mutation operator.
* Figure 3 seems to be missing confidence intervals.

---

> ### Author Rebuttal · Authors · 2025-07-31
>
> Many thanks for your detailed and helpful review. We are grateful for your kind comments about the clarity and written quality of the paper as well as the significance of the analysis.
>
> **Motivating RLO.** RLO may appear artificial at first, but it is archetypical of games that require retention of knowledge. This motivation is explained in detail in our response to Reviewer 97ie.
>
> **The meaning of “small”.** The results presented are stated in terms of n, which always represents the number of game positions. For example, when we derive polynomial runtimes, this means polynomial in terms of the number of game positions. A key limitation here is that the most challenging and interesting combinatorial games (such as Chess or Go) have a combinatorial explosion in the number of positions, and so a result that is polynomial in the number of positions is not practical. In this sense, we say “small” to refer to so-called model or toy games where the number of positions is sufficiently limited for results expressed in this way to be meaningful. We emphasise (as discussed in lines 40-55) that analysis of coevolution on model games is most remarkable in that it provides insights into averting challenges faced by the algorithms more generally.
>
> **Extending our approach.** Our proof provides a framework which can in principle be extended to other games, and we have discussed this in our response to Reviewer 7KRp. In terms of extending the analysis to games of practical interest, the most important consideration relates to moving beyond the model games described above. For very large and interesting games it is impractical to list an explicit move preference at every position. In such settings players use models to map representations of game positions onto move preferences. The most significant challenge is to find a way to adapt theoretical analysis from the exhaustive representation in this paper to a simple instance of such a model. Nonetheless, even for complex models on practical games, the need for diverse and challenging opponents in multi-agent training has been observed empirically [A, B, C]. Our work is the first to analyse this need theoretically for coevolutionary algorithms, and will play an important role in future analysis on practical games.
>
> We are unsure what is meant by analogous results for EDAs, as the algorithm we analyse is a coevolutionary instance of an EDA (see line 113). If your question concerns how to prove analogous results for a non-coevolutionary EDAs, then our black-box results in Section 6 indicate this is not generally feasible (see lines 301-302).
>
> We believe the statistical significance of the impact of the mutation operator is demonstrated using Welch’s t-test in Table 1, which presents the same data as Figure 2 just restricted to the two versions of Tournament UMDA (which are most critical for comparison). Can you please advise if there is a particular experiment or statistical analysis we could provide that would strengthen the empirical evidence?
>
> Following your advice and similar suggestions from other reviews, confidence intervals will be added to Figure 3. Thank you also for your detailed list of corrections.
>
>
> [A] Vinyals, O., Babuschkin, I., Czarnecki, W.M. et al. Grandmaster level in StarCraft II using multi-agent reinforcement learning. Nature 575, 350–354 (2019).
>
> [B] Ranjeet, T.R., Masek, M., Hingston, P., Lam, CP. (2011). The Effects of Diversity Maintenance on Coevolution for an Intransitive Numbers Problem. In: Wang, D., Reynolds, M. (eds) AI 2011: Advances in Artificial Intelligence. AI 2011. Lecture Notes in Computer Science(), vol 7106. Springer, Berlin, Heidelberg.
>
> [C] Lubberts, A. and Miikkulainen, R., 2001, July. Co-evolving a go-playing neural network. In Genetic and Evolutionary Computation Conference GECCO 2001.

---

### Decision · Program_Chairs · 2025-09-17

**Decision:**

Accept (poster)

**Comment:**

All reviewers recommend acceptance. The general opinion is that this paper is an interesting contribution to an important problem (the notoriously difficult dynamics of competitive coevolution).

Therefore, I recommend acceptance.